# Building a Digital Twin Simulator Checking the Effectiveness of TEG-ICE Integration in Reducing Fuel Consumption Using Spatiotemporal Thermal Filming Handled by Neural Network Technique

**Ahmed M. Abed** [1,2,*] **, Laila F. Seddek** [3] **and Samia Elattar** [4,5]

1 Department of Industrial Engineering, College of Engineering, Prince Sattam Bin Abdulaziz University, Alkharj P.O. Box 16273, Saudi Arabia
2 Industrial Engineering Department, Zagazig University, Zagazig P.O. Box 44519, Egypt
3 Department of Mathematics, College of Science and Humanities in Al-Kharj, Prince Sattam Bin Abdulaziz University, Alkharj P.O. Box 11942, Saudi Arabia
4 Department of Industrial and Systems Engineering, College of Engineering, Princess Nourah Bint Abdulrahman University, Riyadh P.O. Box 11564, Saudi Arabia
5 Department of Industrial Engineering, Alexandria Higher Institute of Engineering and Technology (AIET), Alexandria P.O. Box 21311, Egypt
* Correspondence: a.abed@psau.edu.sa; Tel.: +966-509506811

**Abstract:** Scholars seek to recycle wasted energy to produce electricity by integrating thermoelectric generators (TEGs) with internal combustion engines (ICE), which rely on the electrical conductivity, $\beta$, of the thermal conductor strips. The TEG legs are alloyed from iron, aluminum and copper in a strip shape with specific characteristics that guarantee maximum thermo-electric transformation, which has fluctuated between a uniform, Gaussian, and exponential distribution according to the structure of the alloy. The ICE exhaust and intake gates were chosen as the TEG sides. The digital simulator twin model checks the integration efficiency through two sequential stages, beginning with recording the causes of thermal conductivity failure via filming and extracting their data by neural network procedures in the feed of the second stage, which reveal that the cracks are a major obstacle in reducing the TEG-generated power. Therefore, the interest of the second stage is predicting the cracks' positions, $P_{i,j}$, and their intensity, $Q_P$, based on the ant colony algorithm which recruits imaging data (STTF-NN-ACO) to install the thermal conductors far away from the cracks' positions. The proposed metaheuristic (STTF-NN-ACO) verification shows superiority in the prediction over [Mat-ACO] by 8.2% and boosts the TEGs' efficiency by 32.21%. Moreover, increasing the total generated power by 12.15% and working hours of TEG by 20.39%, reflects reduced fuel consumption by up to 19.63%.

**Keywords:** waste heat recovery; damage detection; non-destructive testing; thermal filming; digital twin; optimization; influencing factors

## 1. Introduction

A hybrid vehicle (HV) is one that draws its power from two or more different energy sources. The fundamental idea behind hybrid cars is that the various motors perform better at various speeds; the internal combustion engine (ICE) performs better at maintaining high speed of the vehicle than a standard electric motor while the electric motor is more effective at providing torque, or turning power. Speeding up and switching from one to the other at the right moment results in a win-win situation for energy efficiency, which increases fuel economy. Therefore, the main objective is to guarantee the integration of the ICE with the thermoelectric generator (TEG) to take advantage of ICE waste heat at the exhaust gate and predicting their reliability. The value of accurate time series significance does not need to be emphasized. There have been decades of studies on this issue, which classified the

solutions approach into two most prevalent methods: the first is statistical (e.g., S-ARIMA, Holt-Winter, Box-Jenkins, etc.) and the other is the Mat-heuristic training approach based on analyzing the time series (e.g., LSTM, CNN, TLNN, and recurrent network models). If there were not two options, each would have advantages over the other. If statistics is followed, the why and what of every result must be explained; while if Mat-heuristic training is followed, findings may be superior but are unwarranted [1]. Therefore, validation is an important stage. This paper intends to design a digital simulator twin to check the usability of waste heat of an internal combustion engine (ICE) for transforming it into electricity efficiently via a thermoelectric generator (TEG). The digital twin relies on TEG and has two plates (i.e., engine gasket), one of both is a hot gasket plate connected to the exhaust gate, while the cold gasket plate is connected to the intake gate through thermal conductor strips. The digital twin was used to test three different alloy textures of gaskets and their legs. The main purpose is to determine the most efficient thermal convection for different gaskets. Therefore, this paper reviews the main causes of heat conduction failure, which are cracks in the hot gasket plate that prevent conductivity by predicting these causes using the metaheuristic approach that helps track these causes and reduce transformation efficiency for electric power. Figure 1 illustrates the cause-and-effect diagram for the failure of thermal conductivity, which cancels the TEG integration.

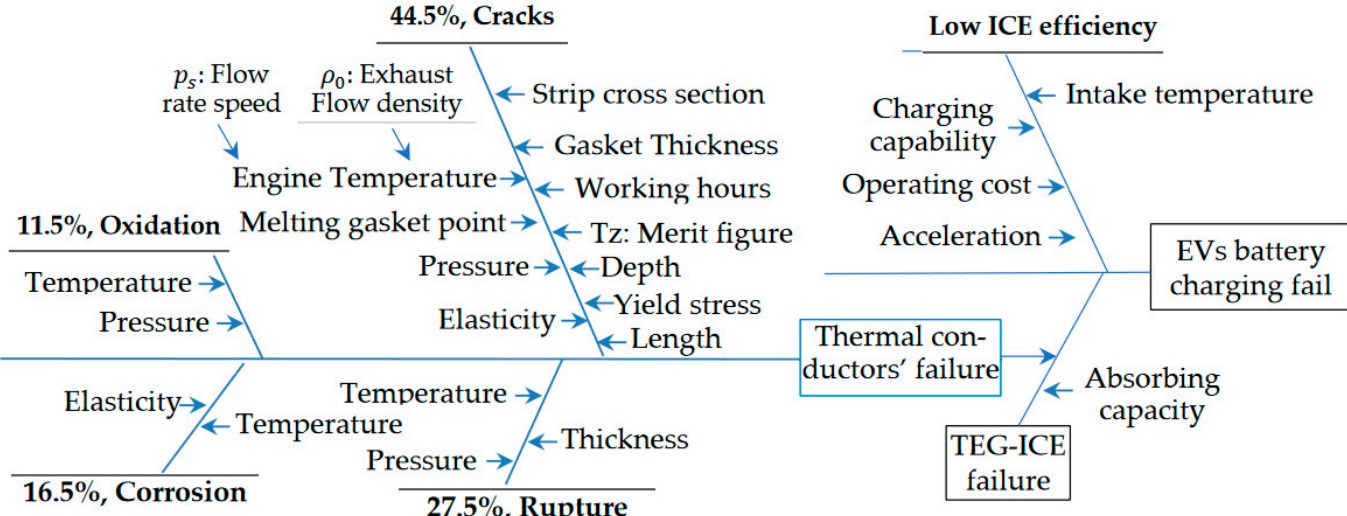

**Figure 1.** The cause and effect diagram for HVs charging failure due to heat conduction failure.

The authors try to predict the cracks' positions via a Mat-heuristic network model to sketch the installation of the thermal conductor's path more securely far away from these cracks. The experimental observations record the causes of cracks using spatiotemporal thermal filming for seeking a crack-free path above the gasket surface for installing the thermal conductors to study the working span life of the integration of TEG and ICE. The digital twin studies the efficiency of integration, the amount of transferred heat and the electric power generated. This paper provides a comprehensive vision of the main technical development of HVs and emerging technologies for their future application in sustaining the battery for a long time. Key technologies regarding batteries, such as charging technology, electric motors, control, and charging infrastructure of HVs are summarized and considered a keystone for our motivation [2–4]. This paper also highlights the technical challenges for the improvement of operating performance via efficiency, reliability, programming of electronic components, heat waste management, and safety of HVs in the coming stages as deduced from Winslow et al. [5]. Therefore, the interest is in using IoT in tracking the HV battery, which is fed by TEG instead of a native generator and studying this approach through the digital twin model. The development of electric vehicles HVs is an urgent necessity, especially after it was proven that an average of 25% of carbon

emissions (CE) in the UK and USA are due to internal combustion engines (ICE), which represent 80% of the change in air characteristics, where two-thirds of the air is saturated with $CO_2$, nitrogen oxide represents a third, and 50% is hydrocarbons, which are known as Greenhouse Gases (GHG) [6,7]. There are ten common components in HHV (auxiliary battery, transformer, generator, electric traction motor, exhaust management system, fuel tank, ICE, electronic unit of power control, thermal system management, transmission), and experts advise not to discharge the traction battery, which leads to spoiling and reducing its lifespan [8,9]. Therefore, this paper suggests studying the replacement of the native generator with the thermoelectric generator, TEG, to take advantage of the heat generated during movement. The TEG is fed from the internal combustion engine and/or from the electric traction engine, to support the power level of the battery, which is at its maximum when also using the brakes repeatedly. The auxiliary battery is used to operate the vehicle before the traction battery is used, while the inverter converts direct current to low voltage alternating current to power the vehicle's accessories [10,11]. It is a major problem for the entire globe to benefit from energy waste in many applications. One of the famous frequent types of energy loss is low-grade thermal energy. Traditional methods for turning this thermal energy into electrical energy are ineffective, difficult, and expensive; however, the approved Seebeck effect is exploited by thermoelectric generators (TEGs), which may convert heat energy directly into electrical energy [12]. Peltier and Thomson's effects are also thermoelectric phenomena used in thermal measurements to achieve reversible thermal and electrical energy transformations and vice versa. The thermoelectric generators (TEG) work on the heat waste extracted from ICE generators or other sources that can absorb thermal energy [13,14] and track their behavior through the digital-twin simulator, which is fed by spatiotemporal thermal analysis films to guarantee steady-state electricity power generation. The main component in the TEG is the thermal conductors' legs, which must be better because of the impact on the generated electricity. Therefore, the authors present a smart prediction methodology for a time of failure to prevent the sudden stop that is considered costly [15]. Some of the heat waste sources and their sudden failure causes were discussed through 2020–2021 [16,17]. Therefore, the authors follow the published examples [18,19] for the importance of tracking and predicting the optimal operating conditions using one or more meta-heuristic optimization algorithms to measure the utilization of TEG accurately. The authors tend to use the STTF measurements fed into the digital twin simulator cloud to predict the efficiency of conductors working by a meta-heuristic algorithm, which integrates the STTF technique and a Cuckoo search to enhance the prediction model (STTF-NN-ACO) and is managed through IoT code inserted in the appendix. The rising source temperature, electrical conductivity, merit figure, melting point/volume, thermal conductor surface area, energy per unit area, gasket texture perpendicular slots', exhaust flow rate, exhaust temperature, intake temperature flow, operating period, and cost per W achieve higher power output and other factors or variables used in the proposed digital-twin Simulator model as cracks' span, depth, and length. The thermoelectric conductors' material was developed by improving the $T_Z$ merit figure for enhancing the generated power, such as a ß-Zn4sp3 semiconductor compound that records a one-way 2.6 $T_Z$. The merit figure improved via seeking new materials formed in specific shape and produced an increase in the electric power at the lowest costs, such as Fe-Al-Cu alloys mixed with Lithium as shown in Table 1.

**Table 1.** The main components of three candidate alloys of TEG texture.

| Alloy Sample | Cu% | Zn | Sn | Ni | Fe | Mn | Al % | Cr % | As | Hv | k W/m K | $\sigma_T$ (MPa) | $\sigma_v$ (MPa) |
|---|---|---|---|---|---|---|---|---|---|---|---|---|---|
| (LiFePO$_4$-Al) $\approx$ (Fe − Al) | —— | Rem. | 1.2 | —— | 0.006 | —— | 4.65 | 20 | —— | 85 | 13.5 | 310 | 105 |
| (Cu-LiFePO$_4$) $\approx$ (Fe − Cu) | 15 | Rem. | 0.001 | 0.004 | 0.7 | 0.001 | 1.9 | —— | 0.012 | 102 | 16.07 | 340 | 180 |
| (LiAlH$_4$-Cu) $\approx$ (Al − Cu) | 77 | Rem. | —— | 0.027 | 0.019 | 0.003 | 15 | —— | 0.037 | 120 | 23.4 | 360 | 230 |

For instance, the cost per W is exclusively determined by energy per unit area and running duration when the fuel cost is minimal or virtually free, such as in waste heat recovery (energy recycling). The scientists thus focused on materials with better power output as opposed to conversion efficiency, such as a rare earth compound, $YbAl_3$ (Density: $\rho = 5.68$ Mg·m$^{-3}$), and other alloys that have a low value but yield energy of at least two times as much as any other material and can function across the temperature range of trash [20]. Therefore, a famous generator company expressed the desire to test the efficiency of the candidate three alloys, which are $(LiFePO_4\text{-Al}) \approx (\text{Fe} - \text{Al})$, $(Cu\text{-}LiFePO_4) \approx (\text{Cu} - \text{Fe})$ and $(LiAlH_4\text{-Cu}) \approx (\text{Al} - \text{Cu})$ to form the thermal conductors' strips and texture of ICE gaskets (Hot TEG plates). The authors selected the ICE generator exhaust gasket and intake gasket textures to be woven as ducts of thermal conductors that perpendicularly end with legs (i.e., the two TEG's plates' thicknesses are 1.63 $\pm$0.1 mm, and 2.15 mm, respectively) as illustrated in Figure 2. The machines based on steady current are subjected to failure when the electricity current rate decreases and seek a compensating source, which suggests the use TEG because of its ability to integrate with ICE generators and is deployed in many companies [21,22]. The authors tested three different types of alloys to determine their merit figure and predict their average malfunction time (i.e., reliability). For the first alloy (Fe-Al) the thermal properties of the terminals' alloy iron- chromium-aluminum fitted to thermal conduct propensity for cold brittleness if working at temperatures exceeding 1000 °C with chemical composition is 4.65% Al, 20% Cr, and balanced Fe, whose thermal conductivity is 13.5 W/m K. The second is iron-copper (i.e., Fe-Cu, called 3003, with 0.15% Cu and 0.7% Fe). The third is aluminum-copper (i.e., Al-Cu, with 15% Al with a grain size of ~47 μm) [23–25]. The digital-twin simulator model accurately represents the contact plates (gaskets' texture) and their joined legs with exhaust and intake flow points [26]. The challenge is to monitor the gasket's efficiency without stopping the generator [27–30]. Therefore, the authors suggest predicting the functional change of the gasket texture caused by cracks appearing by spatiotemporal thermal filming, which can feed the meta-heuristic optimization technique supported by mathematical equations to gain accurate prediction results. The tailored digital-twin simulator model [31] consists of an object (TEG and gaskets), a measuring tool (STTF), and a predicting meta-heuristic optimization algorithm, which uses some of the physical parameters related to cracks (e.g., the span, depth, and length in $m \times 10^{-4}$) that is measured by the cracks' center propagation digitally on the gasket surface texture, as discussed by [32,33]. Research is now being performed to overcome the fundamental drawback of thermoelectric generators, which is their relatively low efficiency, and to efficiently use them in a variety of applications for recovering waste heat. The automobile industry produces a significant amount of waste heat as a result of the low braking thermal efficiency of reciprocating engines, which is less than 30% for gasoline engines, according to Hotta et al. [21]. One of the important precautionary measures is not to allow the HVs battery to discharge more than 80% of the capacity so that the state of charge (SOC) does not fall below 20%. This is to protect the battery from over-discharging. This contribution is achieved using thermal conductors of copper, aluminum, and iron alloys between the thermal potential difference points that help generate an electric current that works within a discharge rate within 300 amps to try increasing the lifespan of the battery to an average of 3500 cycles for the Li-ion cells. The cells are 120 W/kg and must be compatible with traction motors that are divided into two types (AC and DC motors) and which are directly connected with the ECM unit to manage the battery in the so-called TEG, BMS, the amount of heat transferred, the voltage generated and refer to the system with voltage temperature monitoring (VTM) [34–36]. The digital twin is specially designed to be a core to improve HVs in reducing fuel consumption.

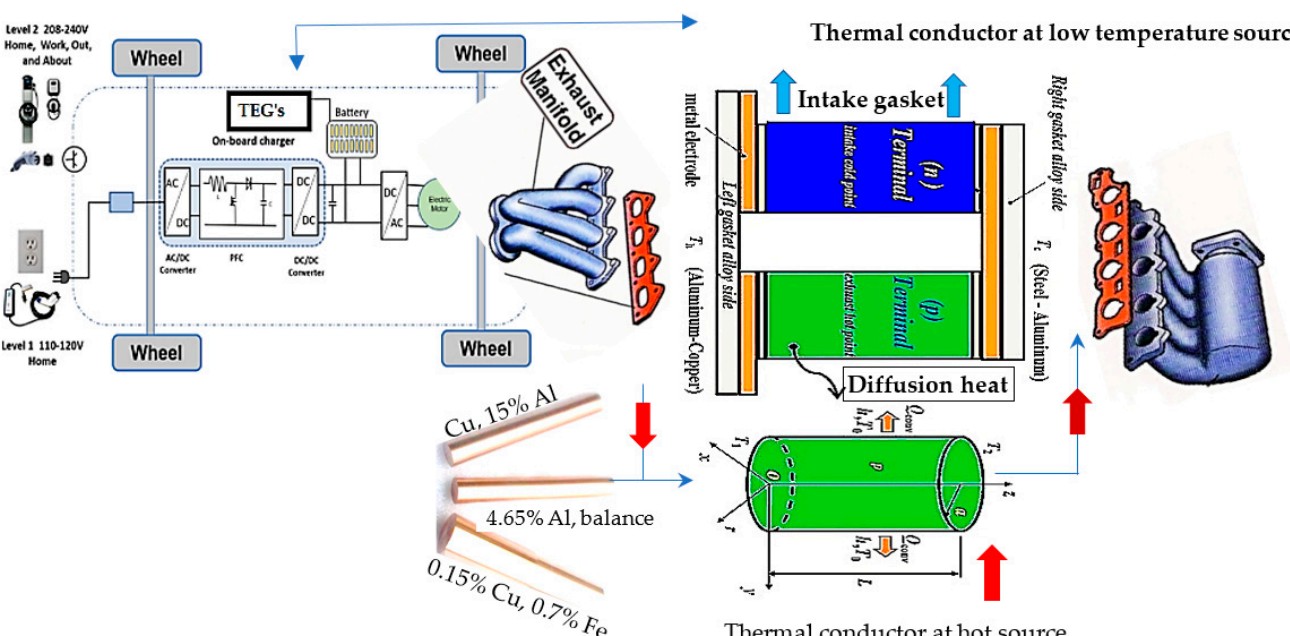

**Figure 2.** The components of the engine under study and TEG according to Dandan Pang, et al., (2022) [37].

## 2. The Architecture of the Digital-Twin Simulators' Paradigm

The exhaust temperature is the hot source useful for TEGs and allows them to convert waste heat into usable electric energy according to [38,39]. Figure 3 illustrates digital twin components and discusses the sequential two stages based on integrating the source [1] with TEG plates. The first stage aims to aggregate the physical measurement parameters [4], which is measured by specific means [3] to test the legs' behavior toward resist failures through crack generation and is named figuratively (reliability stage), to be fed for the prediction methodology [5] which tests the effectiveness of the thermo-conductor's leg material alloy [2] through the second stage. The two stages discussed in Algorithm 1.

---

**Algorithm 1: Main Pseudocode of STTF-NN-ACO mechanism**

---

//The components of the digital simulator's twin (ICE, TEG, and thermal conductor's strips)
//The mechanical and composite of tested alloys discussed by Admiral et al. [40], and Richard & Hertzberg [41]
*//Stage (1): Check the reliability of the thermal conductors by imaging*
*Determine* the TEG gasket plate's install (hot at the exhaust gate, cold at the intake gate)
*Determine* the gasket texture $((Fe - Al),(Fe - Cu),(Al - Cu))$
*for* alloy 1:3
    *if* (gasket texture and its strips from $((Fe - Al))$
        *for* 1 : causes
*Select* the high failure cause (Figure 1: Cause-and-effect diagram);
*Determine* the gasket surface coordinates (*i* denotes the *x*-axis and the *j* denotes *y*-axis);
Using Spatiotemporal Thermal Filming STTF to:
    *Record* the working time (hr.);
    *Locate* thermal conduction failure positions, $P_{(i,j)}^n$, the position of the starting of the cracks' point;
        *for* 1 : cracks
            *Count* the number of each crack's ramifications *b*;
            *Determine* the two longest ramifications lengths for each crack;
            *Determine* the tilt angle of each ramification to the *x*-axis;
            *Determine* the $P_{(i,j)}$ of the end terminal for each tracing ramification;
            *Determine* the intensity $Q_P$ of each crack at specific position (the number of their ramifications);
*//Stage (2): Predicting generated electricity STTF-NN-ACO*
*Predict* the virtual curve line direction of cracks direction as illustrated in (Figure 4) by hybridizing meta-heuristic with the neural network to reduce the error of sketching the secure path of installing the thermal conductors' strips;
        *Measure* the efficiency, *f* (amount of heat conductivity, electrical power generation);
    *Else if* (gasket texture and its strips from next alloy)
    *End*
*End*

---

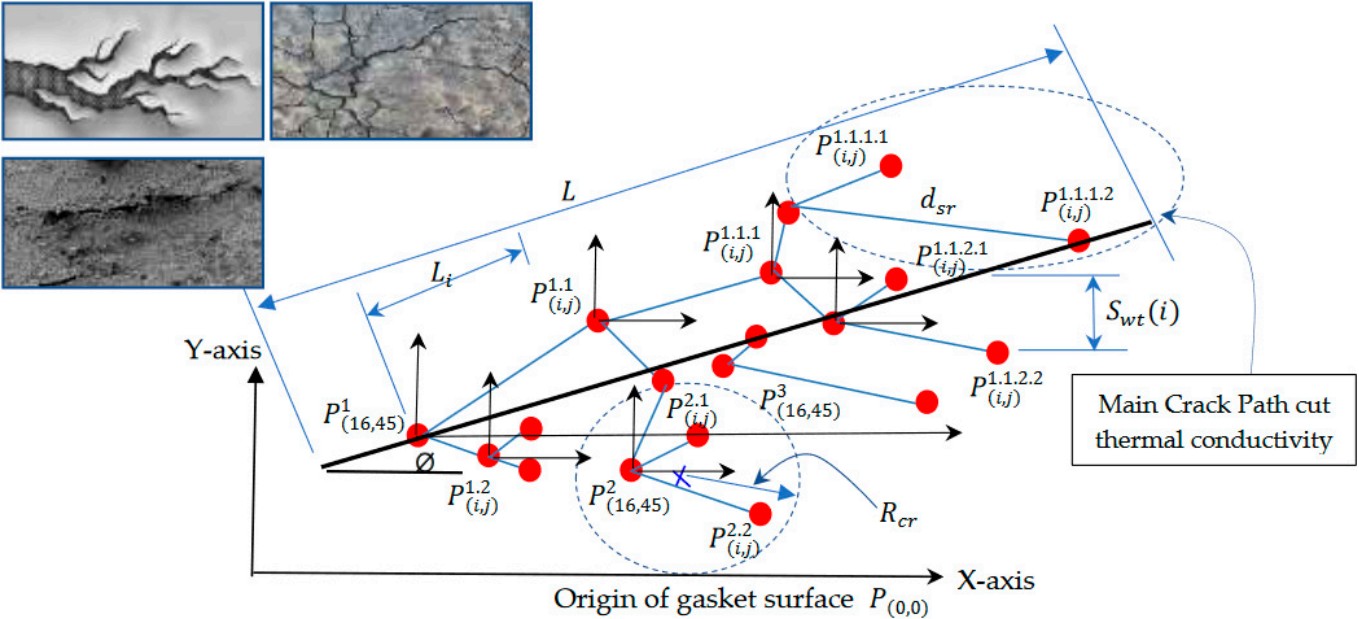

**Figure 3.** The digital twin simulator architecture.

**Figure 4.** A hypothetical visualization of the appearance and growth of cracks and how to track them. ● The crack position; ⟶ represent the coordinates; ⟶ dimension arrows.

The parameters suggested in the proposed mathematical equations are required to form the digital twin simulator for testing the effectiveness of three different alloys (iron, aluminum, and copper) to thermos-conductivity are shown in Table 2. Figure 4 illustrates the sketch of the recording and tracking the crack growth. The pseudocode of the proposed model describes the sequencing of tackling the data input and output to track the efficiency of the TEG-ICE integration.

**Table 2.** The ideal parameters and objective for proposed digital twin model.

| Parameter | Description | Parameter | Description |
|---|---|---|---|
| $D_x, D_y, D_v$ | The coefficient of terminals' cracks diffusion layer type 0.005 mm$^2$ day$^{-1}$ | $\beta$ | The electrical conductivity in the alloy [Sm$^{-1}$] |
| $\rho_0$ | The exhaust flow density [s. mm$^{-3}$] | $\alpha$ | The Seebeck coefficient [V. K$^{-1}$] |
| $p$ | The flow rate speed, mm$^3$ s$^{-1}$ | $\nabla$ | The Hamiltonian operator, |
| $q$ | The dipole source, (0,1) | $T$ | The temperature [K], $0.273 \times 10^3$ slits |
| $k$ | The thermal conductivity [W/m. K] | $b$ | The # of ramifications appeared on the gasket slots 0: 1000 (212) |
| **Spatiotemporal Thermal Filming Parameters** | | | |
| $d_{sr}$ | The distance between two cracks' core on the leg surface [mm] | $L_i$ | The crack segment length among red spots on spatiotemporal images |
| $R_0$ | The radial distance from the gasket surface to picks imaging | $R_{cr}$ | The radius of surrounding circle of terminals' cracks form closed shape. |
| $r$ | The cracks growth rate per week (timespan between two sequential points), 3 weeks | $h_r$ | The radius of the hotspot on the gasket surface, |
| $d$ | The red crack diameter on the spatiotemporal image | $P_{i,j}$ | The cracks' location on the gasket legs' surface |
| Ø | The angle of the cracks line slope line with the *x*-axis | $S_{wt}(i)$ | The cracks' center deviation regression to thermal conductor path according to alloy |
| $Q_p$ | Intensity of the crack is the area of a circle and surrounds all ramifications for specific crack and has position $P_{i,j}$ picked by spatiotemporal imaging, (Dark blue color, Green—Red) | $\omega$ | The relative area of closed perimeter of shared source for cracks by whole gasket area |
| **The digital simulator twin parameters** | | | |
| $c_{fd}$: | The cost of ICE fuel consumption ($0.808 l h$^{-1}$) | $P_w$: | The power generated by ICE (KW) |
| $c_\sigma$: | Underutilization (dereliction) cost ($) | $Q_{df}$: | The ICE consumption, l. h$^{-1}$ |
| $c_d$: | The electricity cost of kW·h$^{-1}$ by a ICE, $/kW·h$^{-1}$ | $f$: | Corrosion rate of alloy layer (mm.day$^{-1}$) |
| $A$ | Cross-section area | $V$: | Crack path length speed (mm/day) |
| $t_m$: | Generator uptime running (wk.) | $K_{i \in (1,2,3)}$: | Coefficients with constant values based on alloy type |
| $t_s$: | The red hotspot area imaging by STTF | $t_{tc}^h$: | The temperature losses from the hot gasket |
| $d_{p,q}$: | Diameter of the hotspot crack leg (mm) | $t_{tc}^c$: | The temperature losses from the cold gasket |
| $g_s$: | Slenderness ratio of gasket layer thickness tolerance ±0.14 mm | $Z\overline{T}$: | The thermoelectric material's performance index $= \alpha_P^2 \overline{T}/\sigma_P \lambda_P$ And $\overline{T}$ and the average temperature of thermoelectric terminal conductors. |
| $P_{min}$: | The minimum absorption, kW | $T_u$: | The downtime due to replacement (hr.) |
| $L_i$: | Crack length mm → (oil spot) | $E(\xi)$: | The exponential distribution with rate ξ |
| $\lambda_p, \sigma_p$: | The heat and electrical conductivity for the leg has type p, respectively. | $\alpha_h$: | The Seebeck coefficient of the hot terminal |
| $T(r, z)$: | The temperature distribution in (r, z) plane | $\delta$: | The standard deviation from average power, kW. |
| $V_{opt}$ | Optimum value of crack path speed (µm/wk.) | $m$: | Total TEG-ICE generator running hours |
| $I$: | The electric current | $z_n$: | Number of red spots on the gasket layer |
| $q_{pconv}$ (a, z): | The heat flow density at the gasket surface of p-type leg $= h[T(a, z) - T_0]$ | $c_t$: | The gasket damage cost ($) (head or intake and exhaust) before analysis expected |
| **Self ACO recruiting parameters** | | | |
| $P_{i,j}^{nl}(t)$ | The position where ants found food | $\eta_{ij}$ | The visibility provides valuable information |
| $N_i^{nl}$ | represents the area that has not been assigned | $\alpha, \beta, \rho$ | The pheromone trail evaporation rate, 4,1,0.8917 |
| $\Delta \tau^{nl}$ | Refers to the pheromone increment ≈ 0.0501 | $EAC_{ij}$, $EAC_{jk}$ | The electrothermal transformation failure cost |
| $nl_s(t)$ | The ants that are planned to move to collect food from different places | $\vartheta$ | A binary parameter (0,1) illustrates the importance of the period of cycle time |
| **Responses** | | | |
| $P_{watt}$ | Generated electrical power of the generator by K-W/hr. | | |
| **Fuel** | The fuel consumption per working week | | |
| $\eta$ | The efficiency of the proposed integration based on heat transfer | | |
| $Di_r$ | The diffusion rate over time per month for each micrometer length | | |

The crack intensity $Q_p$ in our calculation $\leq$ ($Q_{min} = 0$, $Q_{max} = 3$) is determined by the spatiotemporal filming for terminals' cracks. The thermal's pitch strength follows three distribution types as discussed in "Source characteristics of thermal filming." The thermal conductors must be installed far away at these positions, ($P_{i,j}$). The following approximations are proposed in order to streamline the process and guarantee correct simulation results. Ignore the crack paths' and treat it as a continuous straight line projection

parallel to the nearest axis by angle $\emptyset$, $\mu = 0.005$ and, $\sigma^2 = 0.001$ [41,42]. The failure is achieved according to Equation (1):

$$\sum_{P_{1,1}}^{P_{i,j}} Q_P \geq S_{wt(i)} \; \forall i \tag{1}$$

The tracking also reveals that the cracks segment line discussed in Figure 4 above has length L, N ($\mu, \sigma^2, \xi$), and $M_i$ is ranged as $0 < M_i < L$. When the spatial image could not expect the behavior of the positions path or its intensity distribution is not considered, Equation (2) can be followed:

$$S_{wt} = 10 \, log \left[ \sum 10^{0.1(L_i)} \right] \dots \tag{2}$$

The digital twin simulator architecture is sketched in the 2nd section through Figures 2–4 and tests the thermal conductors for electricity via two sequential stages. Tracking the failure causes in three different phases begins with hot gasket, strips' TEG legs, and cold gasket alloys to determine the significant factors to save them from failure in Section 3 (stage-1), while the methodology that relies on the neural network model, which is used to extract the data from STTF to support the ACO metaheuristic technique used to predict the functional change caused by crack diffusion to predict the cracks' position $P_{i,j}$ and their intensity $Q_P$ as discussed in Section 3.1. The mathematical equations are used to enhance the digital twin behavior and NN output data parameters in checking the validity of TEG-ICE integration. Therefore, the data extracted from STTF is considered an initial value of searching using the meta-heuristic optimization algorithm to obtain the optimal results as discussed in Section 3.2, and to determine the battery used to receive generated electricity to serve the HVs sector. Finally, in the list validation in the conclusion discussed in Section 5, the authors suggest using other alloys that have a YbAl$_3$ compound in manufacturing the thermal conductors in the future work section.

### 3. Stage (1): Tracking Thermal Conductors' Failure Causes

This work was a consultant mission for one of the Egyptian generator companies, through the USCC office of Zagazig University, which provides the spatiotemporal thermal filming for the TEG legs through extensive experimental observations, and accumulating data are described as a controlled tracking dashboard. In view of the heat generated by the ICE at the exhaust duct and the movement of the rotating wheels during the use of the brakes at the air duct to generate a difference in the thermal effort and take advantage of the heat and convert it into an electric current by an electric generator and feed the traction batteries to operate the electric traction engine through an electronic control unit (ECM) to regulate the flow of electric power to control engine speed and torque by the transmission via IoT. This is considered very useful in the battery without discharge, as they depend on the TEG in a reciprocal manner by absorbing heat and converting it into electricity to maintain the efficiency of the movement generated when relying on the battery to transition to the mechanical transmission when speeds are less than 40 km/h. The dependence of HVs to begin working on the battery and the electric motor is a reason to save spent gasoline or ICE and reduce CE by an average of 15%. Traction batteries can be evaluated by five influencing factors: specific energy (Wh/kg), acceleration (W/kg), operating cost (cost/km/passenger), fast recharging capability (80% in 10 min), and capacity to absorb high currents during repeated braking [42,43]. The authors have used the thermal filming measurement that tracks the defective place on the studied surface as illustrated in Figures 5–8, which tracks each failure behavior for a specific alloy of strips of TEG legs and hot gasket texture. Figure 5 illustrates the behavior of (Fe-Al) in resisting the crack creation, $P_{i,j}$, which creates ramifications that are discrete around the crack centers uniformly, over $t_m$: 720 working hours, then coalesces and reduces the efficiency toward failure after $t_m$: 1440 working hours.

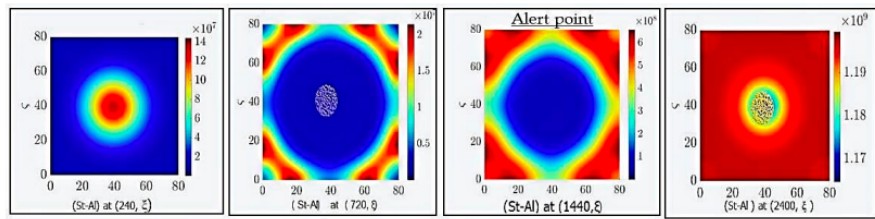

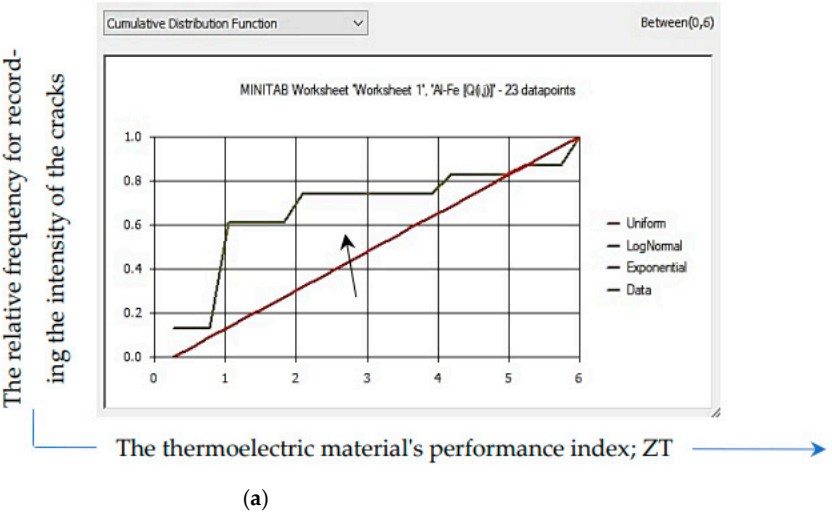

**Figure 5.** The change in the (Al-Fe) gasket alloy resistance working conditions for thermal transfer. (**a**) The behavior of the cracks' growth through 23 weeks for the Fe-Al alloy is uniform.

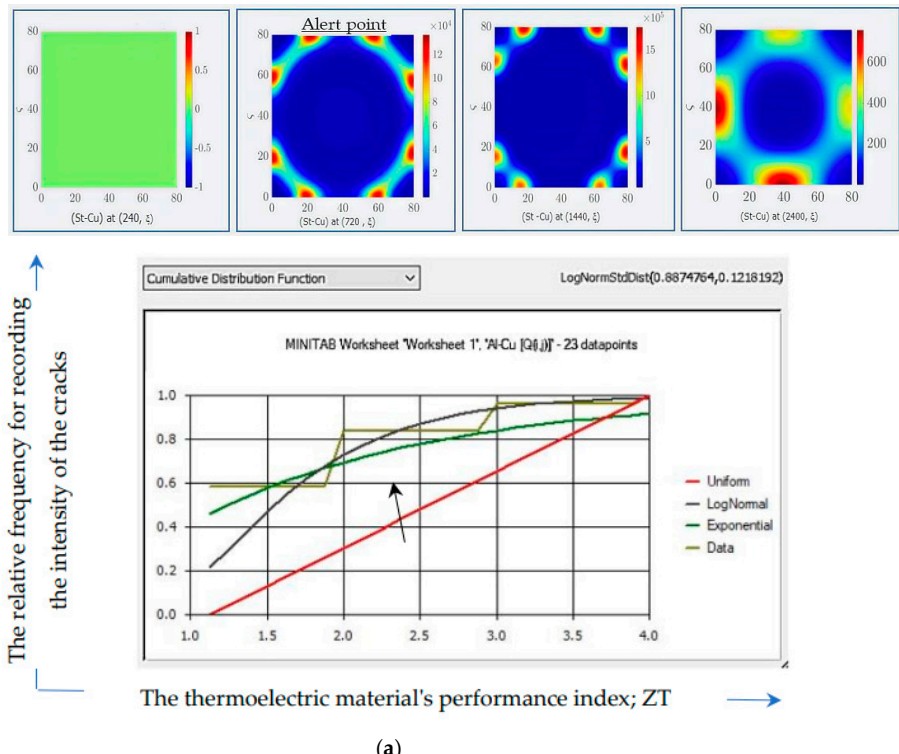

**Figure 6.** The change in the (Cu–Fe) gasket alloy resistance working condition for thermal transfer. (**a**) The behavior of the cracks' growth through 23 weeks for Al–Cu alloy is Gaussian. The arrow points to both lognormal and exponential which have low error than uniform.

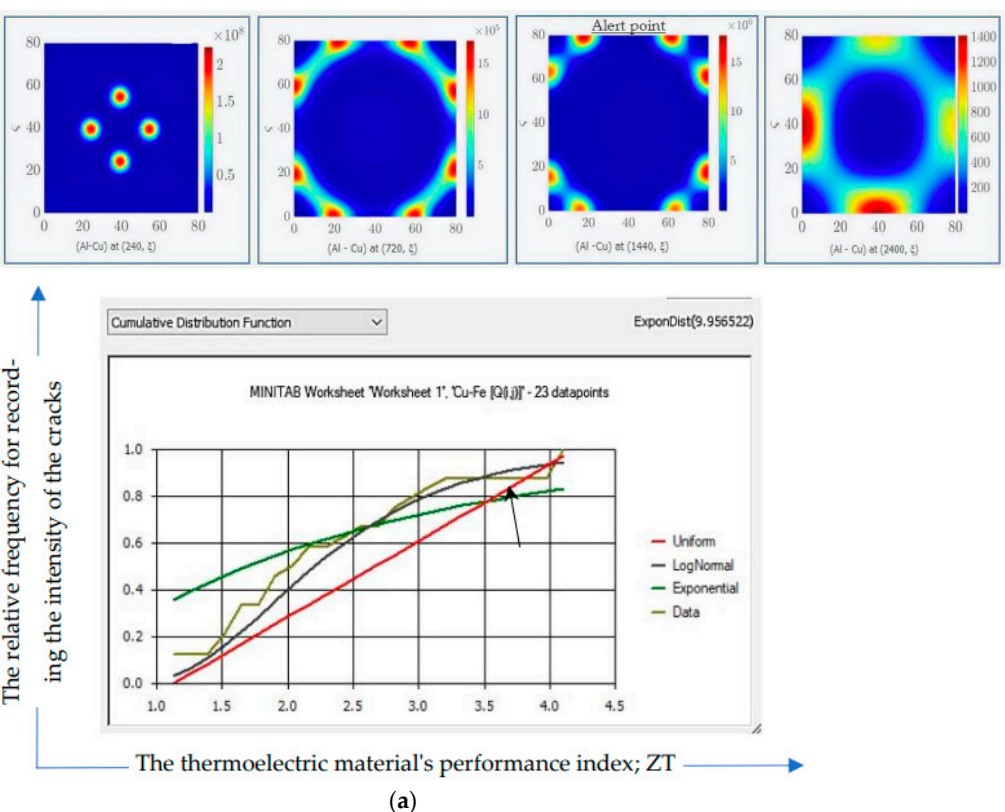

**Figure 7.** The change in the gasket (Al–Cu) alloy resistance working conditions for thermal transfer. (**a**) The behavior of the cracks' growth through 23 weeks for Cu–Fe alloy is exponential.

The cracks and their ramification behavior for the strengthening (Fe-Al) alloy for the gasket and TEG legs to save the thermal conductors via $P_i-$ and $Q_P$-like uniform distributions as illustrated in Figure 5a and formulated as follows: $\begin{cases} P_{i,j} \sim U(0, L) \\ Q_P \sim U(Q_{min}, Q_{max}) \end{cases} \forall i, j = 1, 2, 3, \ldots, n \left(\text{mm} \times 10^{-1}\right)$ The conductor's failure begins in the middle and growth [between (1, 4.1)].

Figure 6 illustrates the (Cu-Fe) alloy after $t_m$: 720 working hours, which resists the terminals' cracks more than (Al-Fe), where the surface damaged slowly in Gaussian behavior than for the (Al-Cu) surface over $t_m$: 1440 working hours, which behaves exponentially as illustrated in Figure 7.

The cracks and their ramification behavior for strengthening the (Cu-Fe) alloy for the gasket to save the thermal conductors via $P_{i,j}$-like uniform distributions, and $Q_P$ is a Gaussian distribution, as illustrated in Figure 6a and formulated as follows: $\begin{cases} P_{ij} \sim U(0, L) \\ Q_P \sim U(\mu, \sigma^2) \end{cases} \forall i, j = 1, 2, 3, \ldots, n \left(\text{mm} \times 10^{-1}\right)$. The conductor's failure forming the surrounding circle is indicated in the following expression: [LogNormStdDist (0.7969939, 0.3705614)]. While the crack and their ramification behavior for strengthening the (Al-Cu) alloy for the gasket and TEG legs to save the thermal conductors is fixed for $P_{i,j}$, while the $Q_P$ is exponentially distributed, as illustrated in Figure 7a and formulated as follows: $\begin{cases} P_{ij} \sim M_i \\ Q_P \sim E(\lambda) \end{cases} \forall i, j = 1, 2, 3, \ldots, n \left(\text{mm} \times 10^{-1}\right)$. The conductor's failure creates four neighboring spots with following expression: [ExponDist (2.36975)].

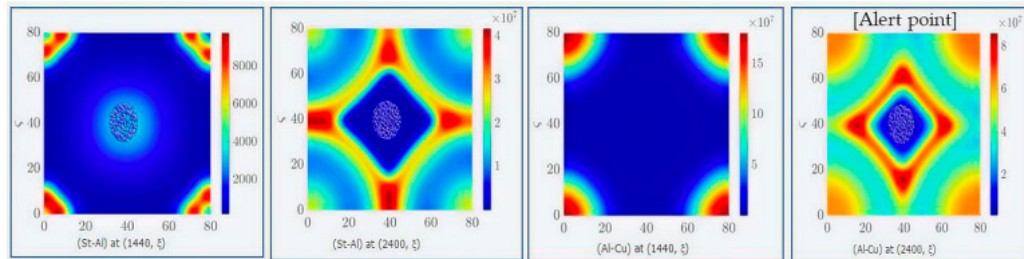

**Figure 8.** The change in the (Fe-Al) and (Al-Cu) alloys resistance working condition for thermal transfer.

Therefore, the authors' experiment uses a gasket lined on both sides, (Al-Fe) on the upper face and (Al-Cu) on the lower face. The authors tend to increase the thickness of the gasket alloy layer to 1.74 mm. The spatiotemporal evolution tracks the gasket terminals' cracks for the (Al-Fe) and the (Al-Cu) as illustrated in Figure 8, during $t$ = (1440; 2400) working hours when $R_0 > 1$ ($51 < b < 100$), for which the ramifications increased to 100. Although the thickness increased, the films show the cracks exacerbate according to the ramification regression as illustrated in Figure 8, but resists for more than 2400 working hours, with $Dx = Dy = Dv = 0.2$.

The authors recommend fixing the thermal conductors away from colored positions (e.g., yellow and orange) that appear via STTF, which avoids the terminals' cracks over ***tm***: 4392 working hours for (Al-Cu) and tm: 5184 working hours for the (Al-Fe) alloy.

*3.1. Setting the Significant Parameters of Thermal Conductivity*

The previous step in stage (1) was executed in the laboratory by tracking all significant causes that affect heat transmission effectively and reflect positively on generated power electricity over 23 weeks by analyzing the gasket case every 3 days (i.e., 45 gaskets' filming) for designing the suitable digital twin which can check the success of the integration of TEG-ICE. The second step in stage (1) will interest in extracting images' data by neural network supported by mathematical procedures such as the cracks' positions and their intensity precisely to study the generated electricity to charge HVs' batteries for a long time, which reflects positively in reducing fuel consumption and carbon emissions. The neural network will feed metaheuristic techniques such as ACO. The neural network was supplied with STTF images of the reasons for thermal conductivity failure after conversion to grayscale, with each pixel being divided by 256 as binary 0 to 1. After the original picture was reduced to 50-50 pixels, the number of input neurons was substantially decreased to just around two, as shown in Table 3. Analytical recording for the temperature distribution via the digital-twin simulator model using a STTF to heat effect is captured as illustrated in the merit Figures 9–11 and Figures 12–16 to seek efficiency about suggested cross sections of TEG legs, which also discusses the TEG-ICE system sketch and illustrates the coordinates of the heat stream. This study aims to attain the highest utilization rate for the electric generator operations (generated power, working days, and profit), in Equations (6)–(11).

**Table 3.** The neural network tunes.

|  | Parameters | Down | Up |
|---|---|---|---|
| $L_1$ | Neuron number | 2 | 17 |
| $L_2$ | Learning rate | 0.012 | 0.39 |
| $L_3$ | Training epoch | 210 | 2550 |
| $L_4$ | Momentum constant | 0.11 | 0.95 |
| $L_5$ | Number of training runs | 3 | 7 |

The digital twin is designed to test the legs' behavior toward resist failures as cracks' generation and track the cracks' span, depth, and length and is named figuratively (reliability stage). The crack positions are the points that failed in heat transfer and must be tracked to increase the working period time efficiently by installing the TEG legs far away from the cracks' positions. Analysis of the STTF images illustrated in Figures 5–8 cleared us of the TEG-IEC integration failure caused by crack intensity, $Q_P$, which is proportional to merit figure, temperature, melting point, their cross section, gasket texture plate thickness and their flow rate speed and exhaust flow density as expressed in Equation (3), which affect the thermal conductors which affect electrical conductivity (i.e., TEG legs) within their working time as illustrated in Figures 9 and 10. If these cracks are independently created at a constant average rate, the analogous continuous equation states that the intensity at a particular position meets a negative exponential distribution per unit of time as expressed in Equation (4).

$$\nabla.\left(\frac{\omega}{\rho_0(\nabla p - q)}\right) - p\frac{\rho_0}{\omega} = Q_P \tag{3}$$

$$\beta = \frac{Q_p e^{-kd_{sr}}}{4\pi r} \; \forall i,j \in gasket\ coordinate \quad \Delta\ 0.01\text{m} \times 10^{-4} \tag{4}$$

Figure 9 illustrates that the intake, exhaust temperature, and the merit figure have a significant impact on generated power that can be gained from the conductors molded from (Fe-Al, Al-Cu), while Figure 10 illustrates that the cross section area of the conductor and the melting point have a direct impact on controlling the cracks' span. Figure 11 illustrates the main reason for the diffusions' crack per mm $\times 10^{-2}$ and affect intake temperature and melting point of the candidate alloys.

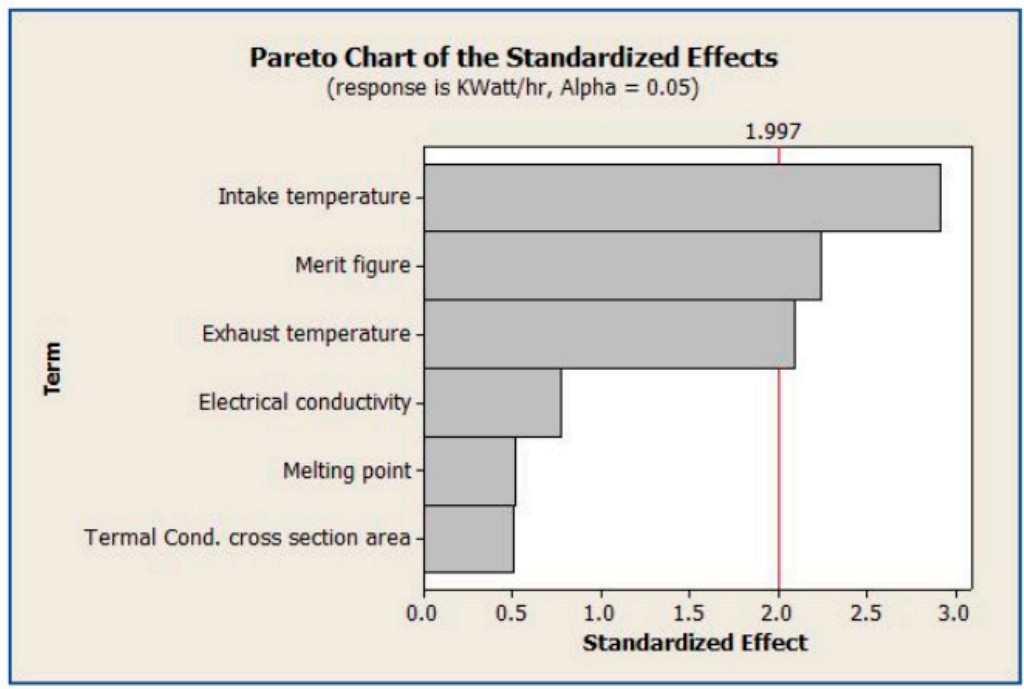

**Figure 9.** The significant parameters affected by power generation.

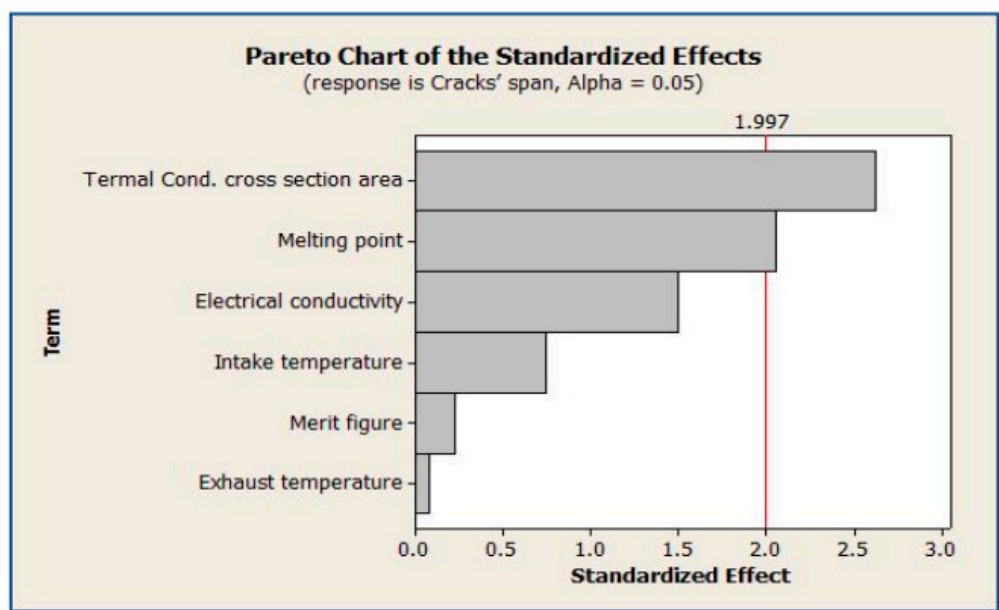

**Figure 10.** The significant parameters affected by power generation and cracks' span.

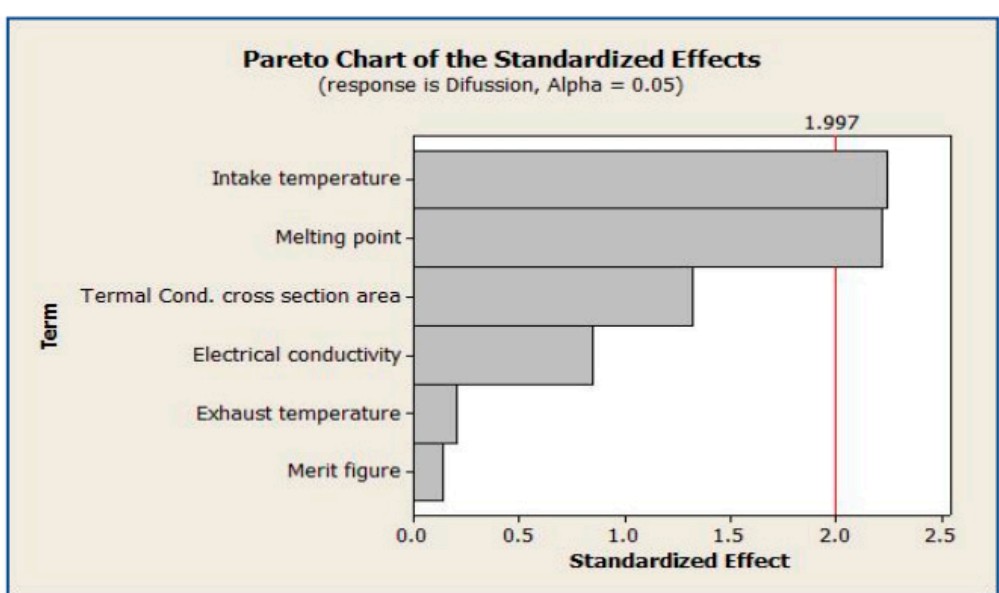

**Figure 11.** The significant parameters affected by the thermal conductor's failure.

The equivalent radius of failure position expressed by $R_{cr}$ in Equation (5), which must install the TEG legs far away from the perimeter of these circles to guarantee electrical power more than 50 W $10^3 \text{hr}^{-1}$, where $R_{cr}$ is the radius of surrounding circle of all cracks positions form closed shape, and $h_r$ is the radius of hotspot ramifications [44,45]. The E ($\xi$) represents the parameter rate $\xi$ for the exponential distribution, while $R_0 < 1$ ($b < 50.0$).

$$R_{cr} = R_0 \left( \frac{bh_r}{R_0} \right)^{\frac{1}{b}} \tag{5}$$

Figure 12 illustrates the final decision for setting the gasket and thermal conductors using conditions and cracks' span and must be controlled in a minimum width of less than 12 mm $\times 10^{-2}$ and the cracks' diffusion are less than five cracks' positions per mm $\times 10^{-2}$. Therefore, the optimizer of the digital-twin simulator indicates the difference between hot

and cold plates and must exceed 105 °C. The merit figure of Al must be 4.3915 times the other materials in the alloys to reduce the cracks' growth for more than 5184 working hours.

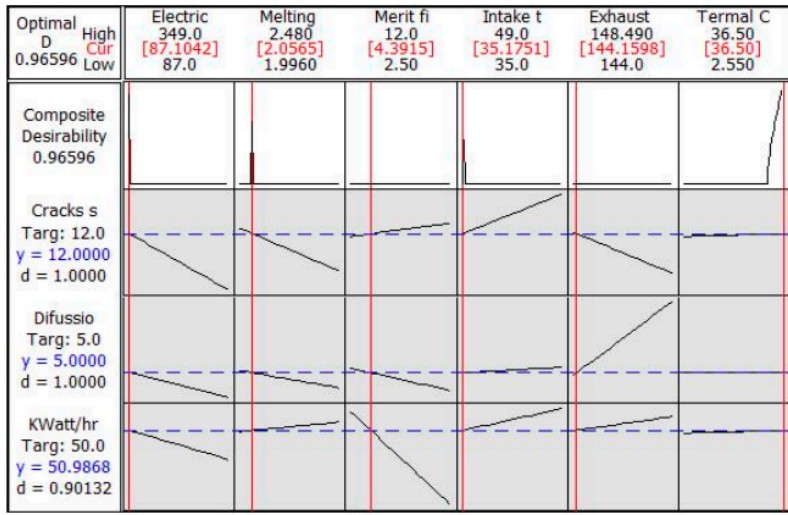

**Figure 12.** The significant parameters affected by the thermal conductor's quick failure on the gasket and terminals strips (TEG legs).

*3.2. Measure the Generated Electricity*

When setting the significant parameters as shown in Figure 12, the authors have divided the heat transmission into three phases. The first phase is through the thermal conductors installed in the hot gasket texture (exhaust flow) far away from the cracks' positions. the second phase is through the TEG legs' strips toward the cold gasket gate. The third phase is in the cold gasket texture. The gasket texture and TEG legs are heavily influenced by temperature, and changes in that temperature have a substantial impact on performance (e.g., merit figure) [46–49]. The authors show the thermal relationships based on the Seebeck coefficient $\alpha$, voltage, and thermal conductivity $\beta$ of both the hot side position (beginning heat source called $\alpha_{hp(i,j)}$), and cold side position (end heat destination called $\alpha_{cp(i,j)}$) in Equations (6)–(11). The first phase of heat transmission is shown in Equations (6) and (7) and the voltage is expected via relation (7a) for the generated electricity from the heat difference between the hot terminal $\alpha_{hp(i,j)}^{1}$ and cold terminal $\alpha_{cp(i,j)}^{1}$ as illustrated in Figure 13, which calculates the number of HVs' battery cells.

$$\alpha_{hp(i,j)} = 5.9214 \times 10^{-13}T^3 - 3.274 \times 10^{-9}T^2 + 2.42 \times 10^{-6}T^1 - 2.744 \times 10^{-4}T^0 \dots \quad (6)$$

$$\alpha_{cp(i,j)} = 1.292 \times 10^{-13}T^3 + 1.074 \times 10^{-9}T^2 - 9.272 \times 10^{-7}T^1 + 8.96 \times 10^{-6}T^0 \dots \quad (7)$$

$$E = LiFePO_4 + 6C \rightarrow LIC_6 + FePO_4 = 3.2\, volt \dots \quad (7a)$$

The second phase of heat transmission through the thermal conductor strip is shown by Equations (8) and (9), which contribute to the voltage required for electrical conductivity and is illustrated in Figure 14. Therefore, the cross section will be designed between [1.14: 1.31] mm.

$$\beta_{hp(i,j)} = 1/(2.25 \times 10^{-14}T^3 - 1.074 \times 10^{-9}T^2 - 9.272 \times 10^{-7}T^1 + 8.96 \times 10^{-6}T^0) \dots \quad (8)$$

$$\beta_{cp(i,j)} = 1/(-1.25 \times 10^{-14}T^3 - 6.43 \times 10^{-11}T^2 + 9.1 \times 10^{-8}T^1 - 1.06 \times 10^{-5}T^0) \dots \quad (9)$$

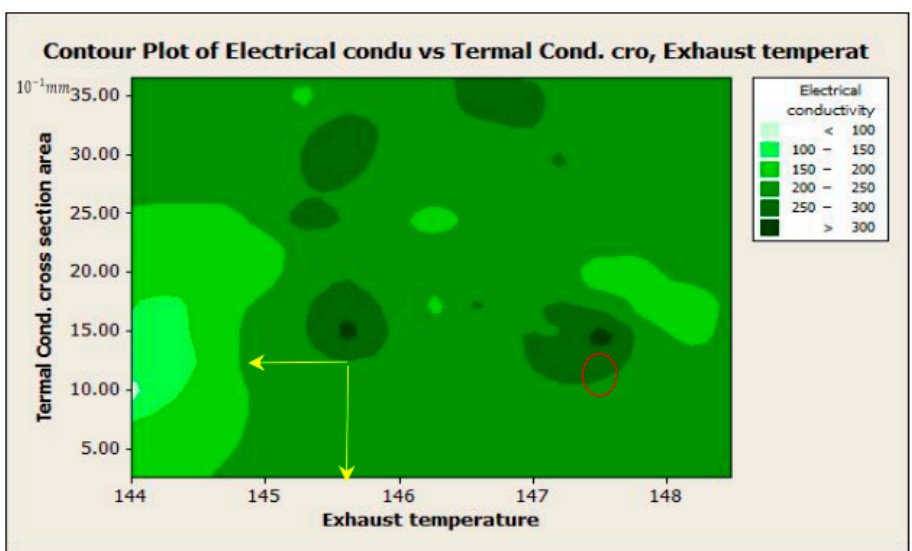

**Figure 13.** The exhaust gasket thickness affects the electrical conductivity at the highest available level. "Hint: the red circle points to the preferred significant parameters values selection".

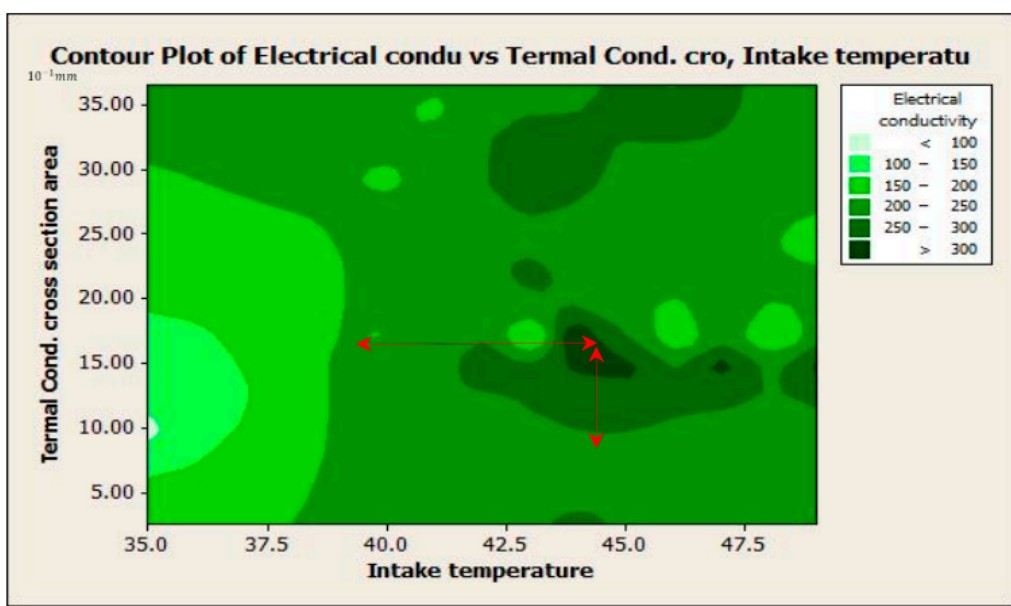

**Figure 14.** The strip cross section according to the intake affects the electrical conductivity at the highest available level. Hint: → points to the preferred significant values.

The cross sections of the TEG legs have a direct impact on electricity affected by the cracks span and their intensity $Q_P$ as illustrated in Figures 15 and 16, which illustrates the cracks' intensity at specific positions to avoid installing the thermal conductors through these positions to guarantee a long life of the power generated. The regression of cracks formed vs. working hours for the different alloys is expounded in the conclusion section.

By the same Equation (7a), computing the voltage affected by the low heat transmission toward a destination side and is shown in Equations (10) and (11).

$$k_{hp(i,j)} = 1.25 \times 10^{-7}T^3 - 1.27 \times 10^{-4}T^2 + 3.87 \times 10^{-2}T^1 - 2.36 \times T^0 \dots \tag{10}$$

$$k_{cp(i,j)} = -1.6 \times 10^{-8}T^3 + 2.91 \times 10^{-5}T^2 - 1.58 \times 10^{-2}T^1 + 3.73 \times T^0 \dots \tag{11}$$

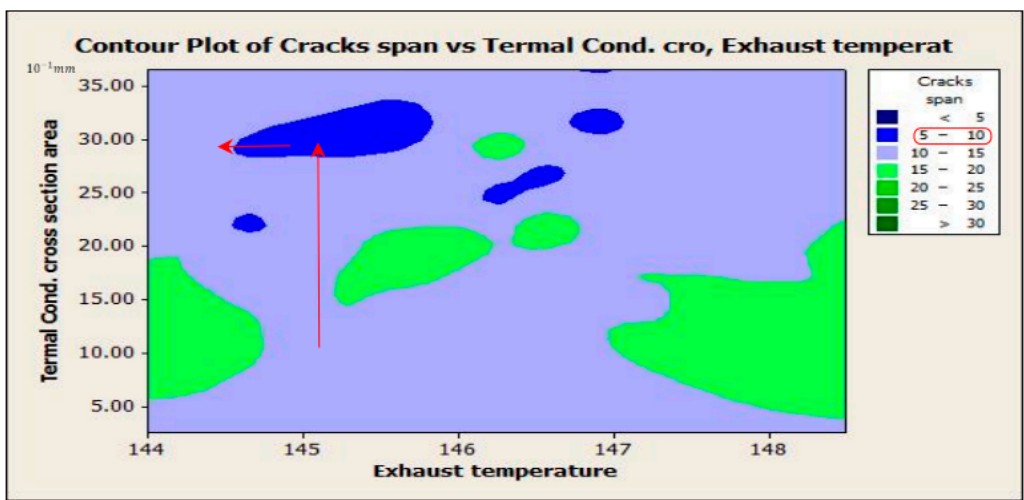

**Figure 15.** The strip cross section for TEG legs affects the cracks' span. Hint: → points to the preferred significant values.

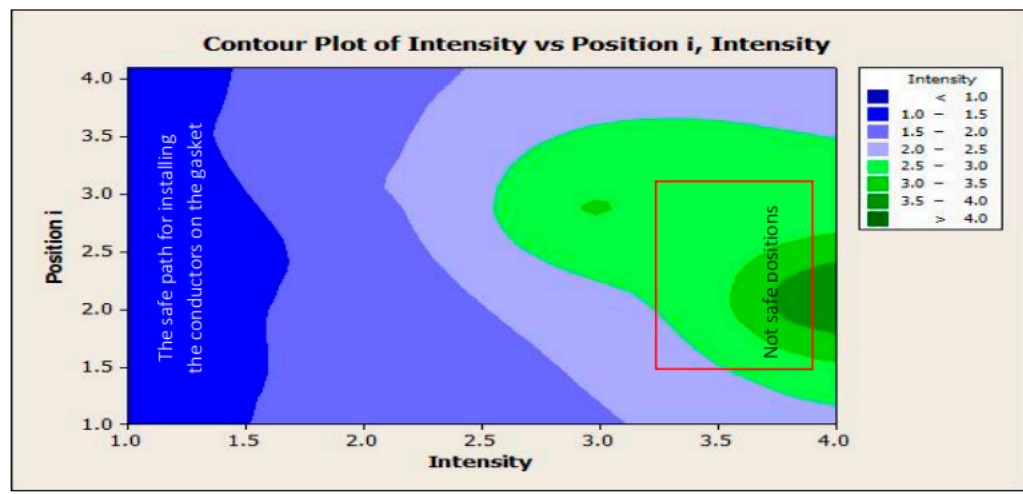

**Figure 16.** The intensity of the cracks' positions for the thermal conductor terminals.

## 4. Stage (2): Predicting the Integration Efficiency

The digital twin works on enhancing the neural network model mechanism by using the advanced artificial ant colony (ACO) setting and relies on data extracted from the analysis of the STTF images. The surface of the gasket was divided into small square areas, each of which has a center that we express by $P_{i,j}$. The main objective is to sustain heat transfer from the hot source (e.g., exhaust gasket) after the thermal saturation case, then allow to direct the heat via strips to arrive at the cold plate (intake gasket). The experimental observations reveal that cracks are the worst cause that hinders the success of the electrothermal transformation of the TEG-ICE system and leads to their integration failure. Therefore, tracking the cracks and installing the TEG legs far away from the cracks guarantees the continuity of electrothermal transformation. The work was divided into two stages. The first stage is discussed above and expounds on the tracking failure causes and their positions by using spatiotemporal thermal filming (STTF). In the second stage, the authors push ants to move to seek food (i.e., cracks' positions; $P_{i,j}$) as expressed in Equation (12) and leave concentrated pheromones equal to their intensities ($Q_P$; numbers of cracks' ramifications; $b$) as discussed in Equation (13). The evaporation was complete if the line connected between the first trigger point and the cracks' positions illustrated in Figure 4 did not create a path over all damaged positions and is expressed by a line fitting error,

$S_{wt}(i)$. The authors seek to declare the cracks' positions precisely to install the TEGs' legs and achieve continuous electrothermal transformation.

$$P_{ij}^{nl}(t) = \begin{cases} \dfrac{[Q_p(t)]^{\alpha} \, [\eta_{ij}(t)]^{\beta}}{\sum_{j=1}^{d} [Q_p(t)]^{\alpha} \, [\eta_{ij}(t)]^{\beta}}, & if \; j \; \in \; N_i^l \leq 20 \\ 0, & otherwise \end{cases} \tag{12}$$

$$\tau_{ijk}(t+1) = (\rho)\tau_{ij}(t) + (\rho)\tau_{jk}(t) + \Delta\tau^l \tag{13}$$

where $Q_P$ is the concentration of pheromone in position $(i, j)$. $N_i^{nl}$, represents the area not assigned. $\eta_{ij}$ The visibility provides valuable information about the problem when searching Dorigo and Gambardella (1997) as cited in Hong et al. [50]. The search parameters, which specify the relative significance of the pheromone trail and heuristic data are: $\alpha = 4$, and $\beta = 1$. Additionally, $\rho = 0.8917$ is the pheromone trail evaporation rate, as shown in the first and second terms of Equation (13), which reflects the local and global updates of the pheromone in working hours $t + 1$ based on the solution of working hours $t$, respectively. The term $\Delta\tau^{nl} \approx 0.0501$ refers to the pheromone increment (the number of connected cracks' positions) and is expressed in Equation (14).

$$\Delta\tau^{nl} = \begin{cases} \rho \times \dfrac{(Z_{Q-max} - Z_{ij})}{(Z_{Q-max} - Z_{Q-min})} + 0.1, & if \; ant'nl' \; discover \; \text{the crack} \\ 0, & \text{otherwise} \end{cases} \tag{14}$$

In each working hour $t$, $N$ ants generate $N$ feasible solutions (monitor all positions that are out of service). $Z_l$ is the solution generated by ant $nl \in \{1, 2, \ldots, N\}$ as cited in [50,51]. $Z_{min} = min(Z_l)$ and $Z_{max} = max(Z_l)$, are the best (TEG generates full electrical power) and worst (failed TEG in electrothermal transforming due to crack spreads) total costs of the solutions generated in working hours $t$, where $Z_{min}$ pheromones and temperature are updated to increase the density of pheromones associated with best solutions and decrease for worse solutions. The transportation cost relies on fuel consumption up to working hours $t$ based on Equation (15), which is used to calculate the visibility ($\eta_{ij}$) for assigning a downstream to an upstream (16).

$$\eta_{ij} = \frac{1}{EAC_{ij}} \tag{15}$$

$$\eta_{jk} = \frac{1}{EAC_{jk}} \tag{16}$$

where $EAC_{ij}$ and $EAC_{jk}$ are the electrothermal transformation failure costs which are proportional to fuel consumption cost and the amount of carbon emissions and are expressed by Equation (17).

$$EAC = G_t \; cost \times avilable \; working \; time + (electrical \; conductivity \; losses \; cost \times working \; uptime) \tag{17}$$

When comparing the best solution of working hours with the best global solution, both are the same in the first working hours. After completing working hours $t$, where $\lambda = nl_s(t)$ is the ants that are planned to move to collect food from different places, $P_{i,j}$, and $\vartheta$ is a binary parameter (0,1), and illustrates the importance of the period of cycle time searching. Divergence is exacerbated by lower values while being reduced by higher values. Equation (18) demonstrates how to enhance the solutions by fading any ant behavior pattern out of the planned pattern. The $\Delta$ is the discrete Laplacian operator, and $\gamma$ is the running time by seconds for ants' acceleration as shown in Figure 17 and illustrates the catalyst that depends on the placement and timing of the actuator's ants. As a result, this may explain

the likelihood that a deviation will occur at the decisive point $(P_{i,j})$ by $\rho(S, t) = \vartheta\gamma^2$. $d_s(t)$ is subtracted from both sides of the equation and then divided by $\delta t$, [52].

$$\rho_s(t + \delta t) = ln\left[(1 - \vartheta)\rho_s(t) + \frac{\vartheta^2}{\lambda}\sum_{S'\sim S}\Delta\alpha_{s'}(t)\right](1 - \omega\delta t) + \beta n_s(t)p_s(t) \qquad (18)$$

$$\rho_s(t + \delta t) = ln\left[\rho_s(t) + \frac{\vartheta\gamma^2}{\lambda}\Delta\rho_s(t)\right](1 - \omega\delta t) + \beta n_s(t)p_s(t) \qquad (19)$$

$$\Delta\rho_s(t) = \frac{\left(\sum_{s'\sim s}\Delta\rho_{s'}(t) - \lambda\rho_s(t)\right)}{\gamma^2} \qquad (20)$$

Take the limit as $\delta t$ and $\rho^2$ both decrease in size, and the ratio $\rho^2/\delta t$ constant remaining with a value specified as $P_{i,j}$ (i.e., the predicted position of food at each phase), and the amount $\beta\delta t$ likewise remaining constant. The suggested equation provides the pheromone-based dynamics of the behavior of finding food sites.

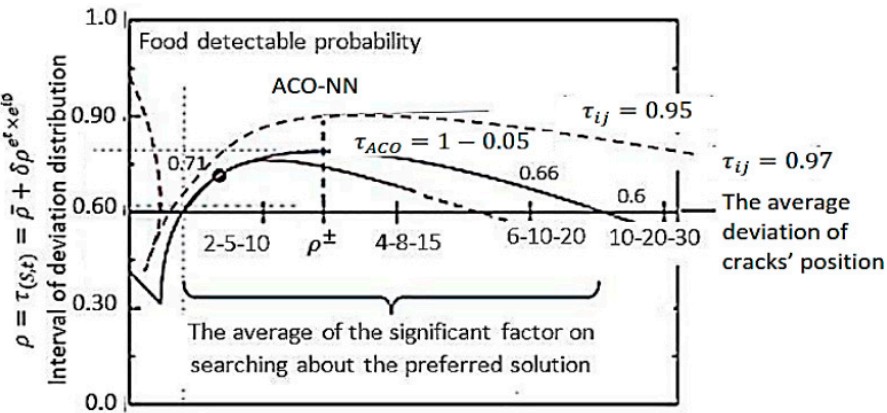

**Figure 17.** The predictable $\rho$ of pheromone in Equations (19) and (20).

The failure permeates the sites, withers (evaporating) over time, and interacts with the ants to encourage them to seek out other locations. Ants react to failure by consistently performing, which causes them to become drained (i.e., inactive), when $Z_{i+1} - Z_i = 0$. $\tau \equiv 1/\omega$ and establishes the timing for the proposed STTF-NN-ACO. Since the failed propagation is either zero or periodic, the fail rate density is matched with the stationary of Equations (19) and (20) which are integrated across the full phase.

The STTF-NN-ACO methodology strengthens the metaheuristic technique with mathematical equations to quickly search for the optimal values for controllable variables discussed in stage (1) to be fed into the digital simulator twin model. Some of the uncontrollable inputs (uncertain) are hard to measure as cited by [53]. Modern simulation resorted to constructing a suitable digital twin for the studied system to be trusted to get the outputs as discussed by Shen et al. [54] and Gan et al. [55] through using spatiotemporal thermal filming which feeds the model continuously [56]. At the same time, with the help of the optimization algorithms, the digital twins' development changes from merely descriptive to becoming actionable. The estimate in the real system resorts to uncertain parameters with their predicted values and may result in a cautious out-put with unnecessarily high operating costs [57]. The stochastic approach was developed by Kalantari (2020) when highlighting the significance of deterministic meaning based on a fuzzy neural network for reducing the error in the searched optimal value [58]. A mathematical optimization enhances the metaheuristic searching by hybridizing with another artificial technique as discussed by Yang et al. and Qiu et al. [59,60]. Stage (1) in this work indicates that if the working parameters are optimized, excellent results are achieved. Therefore, Han et al. discussed the importance of parameter optimization based on metaheuristic methods [61]. SDS has become a robust and efficient global search and optimization technique that was

theoretically defined extensively, especially when combined with other swarm intelligence (SI) algorithms, such as artificial bee colony (ABC) to robust searching techniques jumping over the local optimal solution to global one [62,63]. There is no one optimization technique that is best suited for a wide range of applications or handling diverse sorts of problems. Therefore, the combination and setting of the initial search values for parameters is a very urgent necessity as discussed in stage (1) of this work and is based on Khosravy et al. [64] in analyzing the spatiotemporal thermal filming images for the TEG-ICE component behavior to deduce the preferred working conditions. The challenge is adapting the search mechanism rapidly by predicting specific significant variables as discussed in the cause-and-effect diagram and having the most impact on the results. In the 21st century, most of the current studies adopt an innovative improvement of neural networks by interfering with one of the metaheuristic methods such as the gravitational emulation local search and applied in the electricity-producing optimization in 2010 [65]. In the case of the multi-objective optimization model using the linear decreasing particle swarm optimization algorithm, which is recommended and advised with using the weighted superposition attraction algorithm (WSA) when a solution must be chosen in a minimum time for problems that have multi-pass [66], which was used for the parameter selection and appeared excellent over the native PSO. This thinking approach of the "need-based" must be a guide matched with the understanding of the behavior of the operating parameters through real operating, mainly if applied considering multiple constraints. Therefore, the authors began with laboratory observation by STTF.

### 4.1. The Virtual Suggested TEG Design

The review revealed unanimous in candidate the hybridization between the whale optimization algorithm (WOA) with the fuzzy neural network (FNN), and ACO with GELS (i.e., gravitational emulation local search) to tackle the multi-objective optimization and rapidly gain the global solution [65]. The proposed TEG-ICE mechanism (i.e., digital simulated twin) as illustrated in Figure 18 combines the ACO with NN model which is supported by mathematical equations that are fed data inputs via spatiotemporal thermal analysis through a neural network model to describe the whole behavior of TEG-ICE integration [STTF-NN-ACO] under deduced conditions from the stage (1). The proposed [STTF-NN-ACO] variables are indicated in Table 2. The objective is to accurately predict the crack positions, $P_{i,j}$, and their intensity, $Q_P$, to evaluate the risk of damage level by selecting the best locations sitting on the thermal conductors. These goals are divided into two sub-objectives. The first is tracking the cracks' creation positions, $P_{i,j}$, and their intensity, $Q_P$ [66]. The neural network extracts data images ($P_{i,j}$, $Q_P$, $r$, $\omega$, $h_r$, $R_{cr}$, $R_0$, $d$, , $L_i$, $d_{sr}$) through many iterations approximate to 1000 to obtain less deviation. Therefore, the initial search values rely on significant parameter values extracted via the output of the experiments illustrated in Figure 12, enabling prediction of the optimal solution with minimal error and time accurately. The mathematical equations focus on significant factors cut by the vertical red line and affect conductor lifetime, as illustrated in Figures 9–11. The next equations describe the designed digital twin to enhance the tracking of the TEG-ICE integration. Equation (21) is used as a reference to check efficiency of generated power as discussed by Shen et al. [64].

$$\delta \leq P_w \leq 2.5\, P_{min},\ KW \tag{21}$$

The downtime generated by cracks appearing in gaskets or legs of the proposed TEG-ICE (i.e., failure area) leads to integration failure and can be expressed by Equation (22):

$$T_u = \sum_{i=1}^{m} t_{tc}^h + \sum_{i=1}^{m} t_s K_{1i} V_i\ \ f_i^{-1} + \sum_{i=1}^{m} t_{tc}^c \tag{22}$$

Phases of heat transition: hot gasket, legs' strips,  cold gasket

The influence of the thermal potential difference induced by the gasket alloy texture (Al-Fe) or (Al-Cu) and the temperature 'Th' and 'Tc' between the TEG terminals are equal to

those of the hot source and heat sink. All fields rely exclusively on the coordinates 'r' and 'z' due to the geometry configuration's axial symmetry of the TEG leg and the temperature load. The axisymmetric specific application of steady-state heat flow for the p-type leg is thus [67] suggested in Equations (23)–(27).

$$\frac{\partial^2 T(r,z)}{\partial r^2} + \frac{1}{r}\frac{\partial T(r,z)}{\partial r} + \frac{\partial^2 T(r,z)}{\partial z^2} + \frac{I^2}{\lambda_p A_z^2 (\sigma_p)} = 0 \tag{23}$$

$$q_r(r,z) = -\lambda_p \frac{\partial T(r,z)}{\partial r} \tag{24}$$

$$q_z(r,z) = -\lambda_p \frac{\partial T(r,z)}{\partial z} \tag{25}$$

$$T(r,0) = T_h, \ T(r,L) = T_c \tag{26}$$

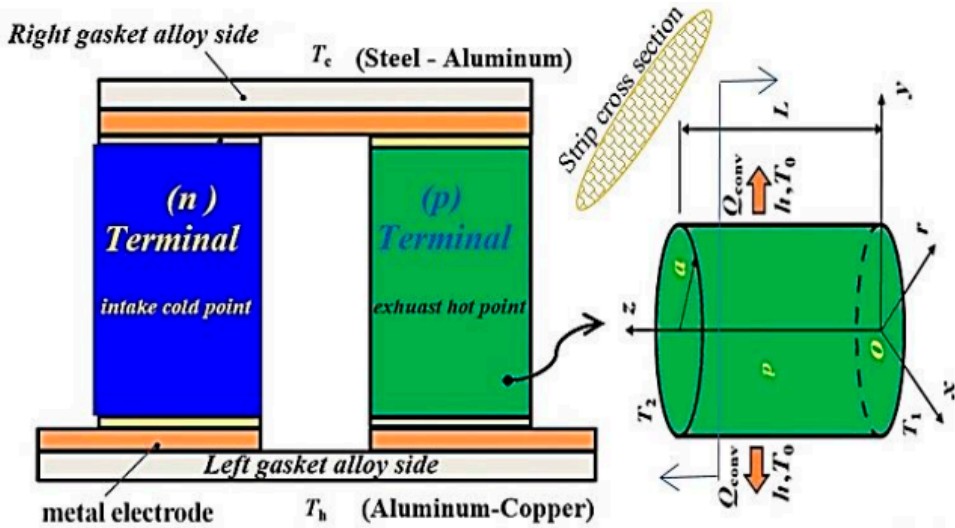

**Figure 18.** A thermoelectric generator with ellipsoid-cross section terminals.

It is assumed that the ambient temperature will always be constant at the maximum value, $T_0$, and the heat convection coefficient between the gasket surface of the p-type leg and its surroundings is *h*. The temperature field's boundary conditions are provided as follows:

$$q_r(a,z) = q_{pconv}(a,z) \tag{27}$$

The solution of Equation (23) for the constraints (26) and (27), using the separation of variables and eigenfunction expansion procedures [68], and is expressed as Equation (28):

$$T(r,z) = -\frac{I^2}{2\sigma_p A_z^2 \lambda_p}z^2 + \left(-\frac{T_h - T_c}{L} + \frac{I^2 L}{2\sigma_p A_z^2 \lambda_p}\right)z + T_h + \sum_{n=1}^{\infty}\frac{h}{\pi}.\Psi I_0(k_b r)\sin(k_b z) \tag{28}$$

where, $\Psi = \frac{[\cos(b\pi)-1]I^2\lambda_p/(\sigma_p)k_n^3 \ L^3+(T_c-T_0)\cos(b\pi)-(T_h-T_0)}{\lambda_p(b\pi)L^{-1}I_1(b\pi L^{-1}a)+hI_0(b\pi L^{-1}a)}$ and consider $I_0(b\pi L^{-1}a)$ *and* $I_1(b\pi L^{-1}a)$ are the zero-order and first-order modifications of the Bessel functions, respectively. Consequently, it is possible to determine the heat transfer rate at the gasket terminal ends as Equations (29) and (30). There is no heat loss between the gasket terminals and the ambient environment.

$$Q_{pz}(0) = \alpha_p IT_h + \int_0^a 2\pi r q_z(r,0)dr = \alpha_p IT_h + K_{pz}(T_h - T_c) - 0.5I^2 R_p - 2\pi a\lambda_p \sum_{n=1}^{\infty} B_b I_1(k_b a) \tag{29}$$

$$Q_{pz}(L) = \alpha_p IT_c + \int_0^a 2\pi r q_z(r,L)dr = \alpha_p IT_c + K_{pz}(T_h - T_c) + 0.5I^2 R_p - 2\pi a\lambda_p \sum_{n=1}^{\infty} B_n I_1(k_b a)\cos(k_b L) \tag{30}$$

The sum of the heat received at the hot source (gasket terminal) $Q_z(0)$, lost by convection at $Q_{conv}$, and dissipated at $Q_Z(L)$ is calculated as expressed in Equations (31)–(33) [38,68]. Where $B_b$ is a constant of the Bessel function that varies according to alloy material, surface to surrounding heat convection, and contact resistance, which has values between 0 to the difference between the hot and cold terminals to go to zero. Additionally, the $Q_{pz}$ and $Q_{nz}$ are the heat transfer rate between positive gasket and negative gasket in the z-axis direction.

$$Q_z(0) = Q_{pz}(0) + Q_{nz}(0) = \alpha I T_h + K_{pz}(T_h - T_c) - 0.5I^2R - 2\pi a\lambda_p \sum_{n=1}^{\infty} B_n I_1(k_b a) \quad (31)$$

$$Q_z(L) = Q_{pz}(L) + Q_{nz}(L) = \alpha I T_C + K_z(T_h - T_c) + 0.5I^2R - 2\pi a\lambda_p \sum_{n=1}^{\infty} B_n I_1(k_n a)\cos(k_b L) \quad (32)$$

$$Q_{conv} = Q_{pconv} + Q_{nconv} = 4\pi ah\left\{ \frac{I^2 R_p L}{12K_{pz}} + \frac{L}{2}(T_h + T_c - 2T_0) - \sum_{n=1}^{\infty} \frac{B_n}{k_n} I_0(k_n r)[\cos(k_b L) - 1] \right\} \quad (33)$$

The power, $P_W$, and the efficiency, $\eta$, of energy conversion from the TEG may be assessed as energy conservation efficiency $\eta$ principle using Equations (34) and (35) as the predicted responses:

$$P = Q_z(0) - Q_z(L) - Q_{conv} = \alpha I(T_h - T_c) - I^2 R \quad (34)$$

$$\eta = \frac{P_{out}}{Q_z(0)}$$
$$= \frac{\alpha I(T_h - T_c) - I^2 R}{\alpha I T_h + k_z(T_h - T_c) - 0.5I^2 R\left(1 + \frac{8h}{aL^2}\sum_{n=1}^{\infty} \frac{I_1(k_b a)[\cos(k_b L) - 1]}{\lambda_p k_b^4 I_1(k_b a) + hk_b^3 I_0(k_b a)}\right) - k_z T_h \frac{4h}{aT_h}\sum_{n=1}^{\infty} \frac{I_1(k_b a)(T_c - T_0)[\cos(k_b L) - (T_h - T_0)]}{k_b \lambda_p k_b^1 I_1(k_b a) + hI_0(k_b a)}} \quad (35)$$

The authors discovered that the heat convection has no impact on the power generated according to Equation (34). The maximum generated power and efficiency is expressed in Equations (36) and (37):

$$I_P = \frac{\alpha(T_h - T_c)}{2R} \quad (36)$$

$$I_\eta = \frac{\alpha(T_h - T_c)}{R(1 + \sqrt{1 + H.Z\overline{T}})} \quad (37)$$

where the influence of heat convection at the change in the level of thermoelectric terminals is measured by the impact factor H, which is calculated by Equation (38), while $\overline{T} = (T_h + T_c)/2$, and treated as dimensionless and approximated to 0.9368 when the heat convection effect is neglected.

$$H = \frac{\frac{(T_h - T_c)}{2\overline{T}}\left[1 - \frac{8h}{aL^2}\left[\sum_{n=1}^{\infty} \frac{I_1(k_b a)[\cos(k_b L) - 1]}{\lambda_p k_n^4 I_1(k_b a) + hk_b^3 I_0(k_b a)}\right](T_h - T_c)\right]}{1 - \left[\frac{4h}{aT_h}\left[\sum_{n=1}^{\infty} \frac{I_1(k_b a)(T_c - T_0)[\cos(k_b L) - (T_h - T_0)]}{k_b \lambda_p k_b^1 I_1(k_b a) + hI_0(k_b a)}\right]T_h\right]} \quad (38)$$

The greatest power efficiency $\eta_{max}$ through the conversion and maximum generated power $P_{max}$ are expressed in Equations (39) and (40), respectively:

$$\eta_{max} = \frac{T_h - T_c}{T_h} \cdot \frac{\sqrt{1 + H.Z\overline{T}} - 1}{\left(\sqrt{1 + H.Z\overline{T}} + \frac{T_C}{T_h}\right) - \frac{8h}{aL^2}\left[\sum_{b=1}^{\infty} \frac{I_1(k_b a)[\cos(k_b L) - 1]}{\lambda_p k_b^4 I_1(k_b a) + hk_b^3 I_0(k_b a)}\right](T_h - T_c)/T_h} \quad (39)$$

$$P_{max} = \frac{\alpha^2(T_h - T_c)^2}{4R} \quad (40)$$

In this paper, a simulated digital-twin model studied the effectiveness of thermal conductor candidates in transferring thermal energy to electricity through the TEG by forming legs to be ellipsoid in a cross section, and maintain the temperature difference between the exhaust gasket plate and cold intake gasket plate up to 105 °C. The heat trans-

fer equation is nonlinear and untraceable because of temperature-dependent properties such the Seebeck coefficient, and electrical and thermal conductivity [69]. The proposed methodology predicts with the positions and the intensity of cracks to avoid the installation of thermal conductors in these positions to guarantee TEG working > $t_m$: 6421 h for the intake and exhaust gaskets as illustrated in Figure 19.

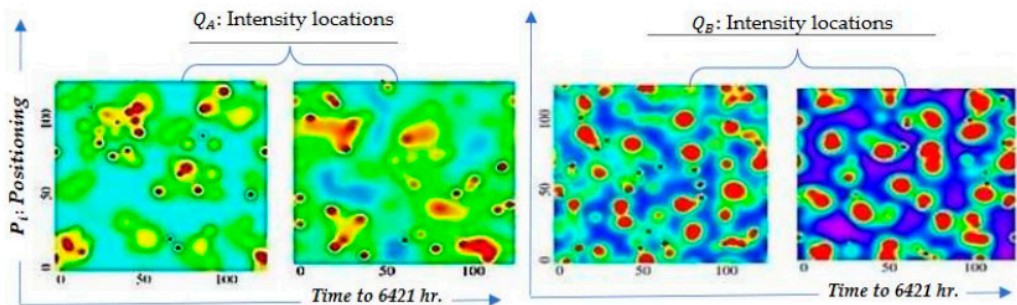

**Figure 19.** The intake (Al-Fe) and the exhaust (Al-Cu) gaskets' conductors before failure.

The predicted efficiency $\eta$ in the case of using the candidate alloy strips and fixed in the safe path as identified by STTF and the proposed STTF-NN-ACO methodology is illustrated in Figure 20a. At the same time, the deviation of the actual trace after releasing the gasket and connected conductors to sketch the failure path thermally (stop condition) has a low value. In contrast, the $\eta$ of electricity generated from the rejected alloy (Cu-Fe) gave the wrong initial values to feed the methodology, resulting in the maximum deviation and a wrong prediction path as illustrated in Figure 20b. The gained power $P_{max}$ by the proposed STTF-NN-ACO when lined by (Al-Fe) and (Al-Cu) sketch a safe path for the conductors far away from the crack positions $P_{i,j}$ and their intensity, $Q_P$.

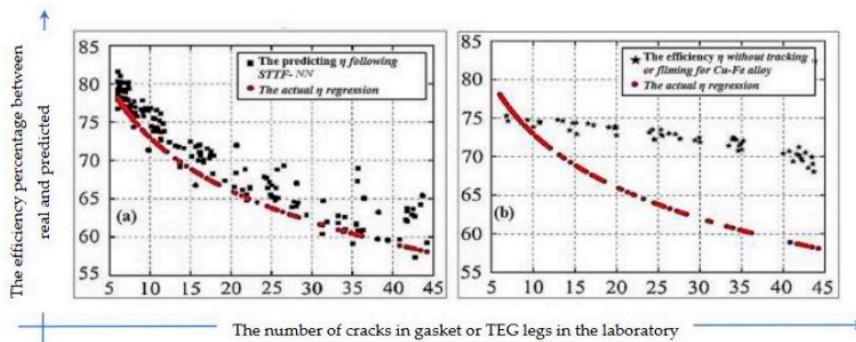

**Figure 20.** Comparison between the digital twin [STTF-NN] utilization results and the exact measurements for the behavior of cracks effects on generator efficiency $\eta$ according to their $P_{i,j}$ and $Q_P$ in two cases: (**a**) Al-Fe and/or Al-Cu alloys; (**b**) rejected Cu-Fe alloy.

*4.2. The Experimental Measurements' to HVs Batteries*

The thermal conductors' strips need to give the required voltage based on the recharging battery specification [69], for instance if the wanted power capacity is 85 KW/hr. Alloyed by Al-Cu with specifications of battery voltage of 350 volt, 3.2 volt for the strip as shown in Equation (7a) and 3.25 Ah. Therefore, the required strip package = 350/3.2 = 110 strips as shown in Equation (7a). The feeding electrical current will be 110 × 3.2 × 3.25 = 1144 W. However, 85 KW hr./1144 = 75 parallel strips will extract 75 × 3.25 = 244 Ah. While using Al-Fe, 108 parallel strips are needed. If an initial efficiency of 25% can save from fossil fuels 0.25 × 12000 = 3000 W per kg of energy used. While in the case of the battery, the efficiency is higher, and therefore 150 × 0.9 = 135 W per kg of energy can be obtained, which increases the battery usage rate 80 times. The cost of the electrical loss due to cracks and conductivity failure $watt.h^{-1}$ can be expressed in Equations (41) and (42) [67–70]:

$$c_d = \frac{Q_{df} \cdot C_{fd}}{P_w} \tag{41}$$

$$
\begin{aligned}
C_\sigma = {} & c_{mat} + (c_\nabla + c_{fd} + c_d + c_t)t_s + \sum_{i=1}^{m}\left(c_\nabla + c_{fd} + c_d + c_t\right)K_{1i}V_i^{-1}f_i^{-1} \\
& + \sum_{i=1}^{m} c_{ti}K_{3i}V_i^{(\frac{1}{b})-1}f_i^{[\frac{\omega+g_s}{b}]-1} + \sum_{i=1}^{m}\left(c_\nabla + c_{fd} + c_d + c_t\right)t_{tci}
\end{aligned}
\tag{42}
$$

The proposed mechanism was evaluated to measure effectiveness by tracking 45 (one exhaust hot gaskets/day) of ICE in the heat lab, studying the fuel consumption, carbon emissions, and generated electric power which is directly reflected on transportation costs as shown in Table 4 and illustrated in Figure 21, and comparing the results with the trial of Hong et al. and Abed et. al. [50,52].

**Table 4.** TEG-IEC fixed and variable-cost ranges.

| A Fixed-Cost $ | Avg. Fuel Cost $+Variable_Cost | Avg. Fuel Cost $+Variable_Cost |
|---|---|---|
| Whole setup experiment parameters | From gasket 1 to 23 | From gasket 24 to 45 |
| (30–50) × Avg. variable-cost | 0.808 + (10:30) | 0.0808 + (10:50) |

The fuel consumption costs are summarized in Table 5 by recording the whole transportation cost as discussed in Hong et al., and Abed, et al. and separating the fuel cost to study the electrothermal transformation efficiency for suitable batteries and calculate the relative percentage enhancing (*RPD*). *RPD* is calculated in Equation (43) and reveals that TEG-ICE integration achieves a 19.63% reduction in fuel consumption.

$$RPD_{ACO-NN} = \frac{Fuel\ Cost_{LINGO} - Fuel\ Cost_{(ACO-NN)}}{Fuel\ Cost_{LINGO}} \times 100 \qquad \forall\, i = 1, 2, 3 \tag{43}$$

**Table 5.** Laboratory observations for gasoline consumption using LINGO and STTF-NN-ACO for transportation cost.

| Number of Gasket /Day | Total Cost $f(Z)$ | | | RPD % | | Number of Gasket /day | Total Cost $f(Z)$ | | | RPD % | |
|---|---|---|---|---|---|---|---|---|---|---|---|
| | LINGO | Mat-ACO | STTF-NN-ACO | Mat-ACO | STTF-NN-ACO | | LINGO | Mat-ACO | STTF-NN-ACO | Mat-ACO | STTF-NN-ACO |
| 1 | 128,524 | 128,524 | 120,812.56 | 0.00% | 6.00% | 24 | 160,355 | 149,618 | 101,590.622 | −6.70% | 36.65% |
| 2 | 114,997 | 114,999 | 104,649.09 | 0.00% | 9.00% | 25 | 137,353 | 137,292 | 106,950.468 | −0.04% | 22.13% |
| 3 | 150,646 | 150,647 | 132,569.36 | 0.00% | 12.00% | 26 | 129,044 | 140,546 | 109,485.334 | 8.91% | 15.16% |
| 4 | 130,997 | 131,006 | 123,145.64 | 0.01% | 5.99% | 27 | 135,362 | 160,355 | 124,916.545 | 18.46% | 7.72% |
| 5 | 121,825 | 123,098 | 113,250.16 | 1.04% | 7.04% | 28 | 101,255 | 137,360 | 111,536.32 | 35.66% | −10.15% |
| 6 | 115,001 | 115,005 | 108,104.7 | 0.00% | 6.00% | 29 | 144,241 | 129,051 | 109,951.452 | −10.53% | 23.77% |
| 7 | 158,396 | 158,359 | 148,857.46 | −0.02% | 6.02% | 30 | 142,535 | 135,369 | 109,919.628 | −5.03% | 22.88% |
| 8 | 157,268 | 158,439 | 141,010.71 | 0.74% | 10.34% | 31 | 101,177 | 101,262 | 81,110.862 | 0.08% | 19.83% |
| 9 | 142,211 | 142,220 | 129,420.2 | 0.01% | 8.99% | 32 | 143,779 | 144,051 | 113,656.239 | 0.19% | 20.95% |
| 10 | 127,857 | 127,877 | 112,915.39 | 0.02% | 11.69% | 33 | 142,535 | 143,014 | 99,394.73 | 0.34% | 30.27% |
| 11 | 122,273 | 122,124 | 100,752.3 | -0.12% | 17.60% | 34 | 132,535 | 132,842 | 91,660.98 | 0.23% | 30.84% |
| 12 | 139,617 | 139,630 | 131,252.2 | 0.01% | 5.99% | 35 | 130,535 | 131,341 | 94,959.543 | 0.62% | 27.25% |
| 13 | 138,248 | 138,264 | 124,299.33 | 0.01% | 10.09% | 36 | 131,535 | 132,044 | 102,730.232 | 0.39% | 21.90% |
| 14 | 150,085 | 150,086 | 136,578.26 | 0.00% | 9.00% | 37 | 177,854 | 178,113 | 131,447.394 | 0.15% | 26.09% |
| 15 | 118,701 | 118,713 | 107,316.55 | 0.01% | 9.59% | 38 | 209,060 | 209,839 | 128,001.79 | 0.37% | 38.77% |
| 16 | 134,079 | 119,207 | 103,710.09 | −11.09% | 22.65% | 39 | 167,758 | 168,004 | 104,162.48 | 0.15% | 37.91% |
| 17 | 157,904 | 118,712 | 104,466.56 | −24.82% | 33.84% | 40 | 179,626 | 179,907 | 105,965.223 | 0.16% | 41.01% |
| 18 | 168,949 | 117,311 | 106,166.45 | −30.56% | 37.16% | 41 | 202,973 | 203,680 | 106,524.64 | 0.35% | 47.52% |
| 19 | 155,656 | 134,093 | 110,894.91 | −13.85% | 28.76% | 42 | 166,785 | 168,884 | 89,508.52 | 1.26% | 46.33% |
| 20 | 170,284 | 157,921 | 153,499.21 | −7.26% | 9.86% | 43 | 182,584 | 185,931 | 106,166.601 | 1.83% | 41.85% |
| 21 | 149,618 | 168,976 | 151,571.47 | 12.94% | −1.31% | 44 | 154,217 | 154,540 | 92,414.92 | 0.21% | 40.07% |
| 22 | 137,285 | 155,663 | 146,323.22 | 13.39% | −6.58% | 45 | 179,461 | 180,016 | 104,949.328 | 0.31% | 41.52% |
| 23 | 140,544 | 170,291 | 149,685.78 | 21.17% | −6.50% | | | | | | |

Since it is more typical to record fuels and lubricants in liters, fuel consumption is also calculated in liters per hour for operating vehicles or autonomous engines and can be expressed in Equation (44) [71].

$$c_{fd} = q_e \cdot N_e / 1000 \cdot \rho_t \qquad (44)$$

where: $q_e$—effective fuel consumption, g . $(\text{kW}^{-1} \cdot \text{h}^{-1}) = (3600 \cdot 10^3) / (\eta_e \cdot H_n)$

$N_e$ — effective power, kW;
$\rho_t$ — gasoline density, 0.76 g/cm$^3$ (kg/Litre);
$\eta_e$ — ICE efficiency = $\eta_i \cdot \eta_m$;
$H_n$ — gasoline calorific value, kJ/kg;
$\eta_i$ — ICE efficiency indicator = $[(P_i \cdot L_0 \cdot R \cdot T) / (H_n \cdot \eta_v \cdot P)] \cdot \alpha$;
$\eta_m$ — mechanical ICE efficiency = $P_e / (P_e + P_m)$;
$P_i$ — regular indicator pressure, kPa = $P_e / \eta_m$;
$P_e$ — regular effective pressure, kPa = $(N_e \cdot 30 \cdot \tau \cdot 10^3) / (V_h \cdot n)$;
$P_m$ — pressure of mechanical losses, kPa = $a_m + b_m \cdot W_n$;
$\tau$ — ICE cycle = 4;
$V_h$ — ICE displacement (all cylinders), l.6;
$L_0$ — stoichiometric amount of gasoline -air mixture, 0.5119 kmol/kg;
$n$ — ICE rotation speed, rpm ($n_{min} = 800 \, rpm$);
R — universal gas constant, = 8.31 J/(mol · K);
T — air temperature, 0.346 K;
$\eta_v$ — ICE cylinder fill ratio = $B_\eta \cdot N_1 + C_\eta$;
$B_\eta$ — empirical coefficients depending on ICE type approximate to 0.17;
P — air pressure, kPa;
$\alpha$ — excess air ratio = $A_\alpha \cdot N_1^2 + B_\alpha \cdot N_1 + C_\alpha$;
$N_1$ — power utilization percentage, % = $(N_e \cdot 100) / N_{e\,max}$;
$A_\alpha$, $B_\alpha$, $C_\alpha$ empirical coefficients depending on ICE type, gasoline ICE are $-1.1^{-4}$, 0.012, 0.85, respectively.
$N_{e\,max}$—maximum effective ICE power, kW;
$W_n$ — average piston speed, m/s = $(30 \cdot S_n) / n$;
$a_m, b_m$ — mechanical loss factors in the engine;
$S_n$ — cylinder height (distance from TDC to BDC), m;

The equation for calculating the hourly fuel consumption in liters per hour is obtained via substitution in Equation (45) [68]:

$$c_{fd} = \frac{0.12 \cdot P \cdot V_n \cdot n}{L_0 \cdot R \cdot T \cdot \tau \cdot \rho_t} \cdot \frac{B_\eta + \frac{10^2 \cdot A_n \cdot N_e}{N_{e\,max}}}{C_\alpha + \frac{10^2 \cdot B_\alpha \cdot N_e}{N_{e\,max}} + \frac{10^4 \cdot A_\alpha \cdot N_e^2}{N_{e\,max}}} = 0.00185 \cdot V_h \cdot \frac{P}{T} \cdot n = 0808 \, Lh^{-1}. \qquad (45)$$

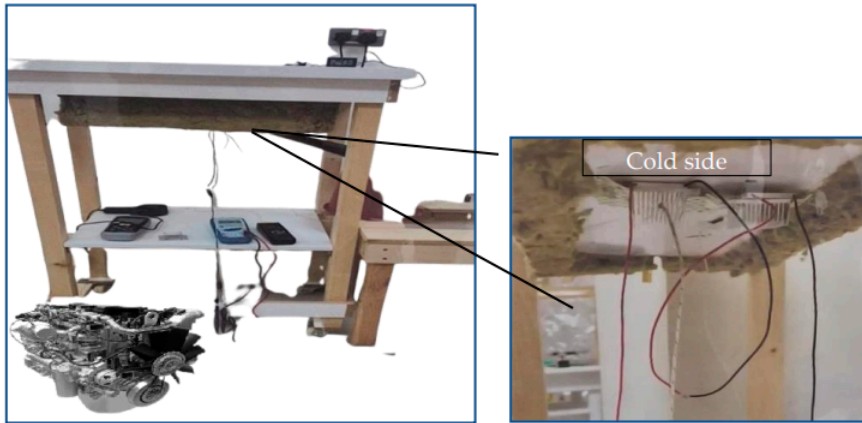

**Figure 21.** The TEG-ICE integration laboratory and observe 45 gaskets through 23 weeks.

Figure 22 illustrates the fuel consumption reflected on generated electricity power (W) and shown in Figure 23, which if reduced as appeared after 5 working weeks, the experiment stops and unfixes the gasket and records the cracks' position and intensity by filming and installing the thermal conductivity strips in another path far away from the direction of the cracks. Therefore, an increase in power is observed until week #9, which is observed as a steady-state behavior to week #20, then reduced again to week #23. The experiment again stops and analyzes all filming shots to discover the cracks' behavior.

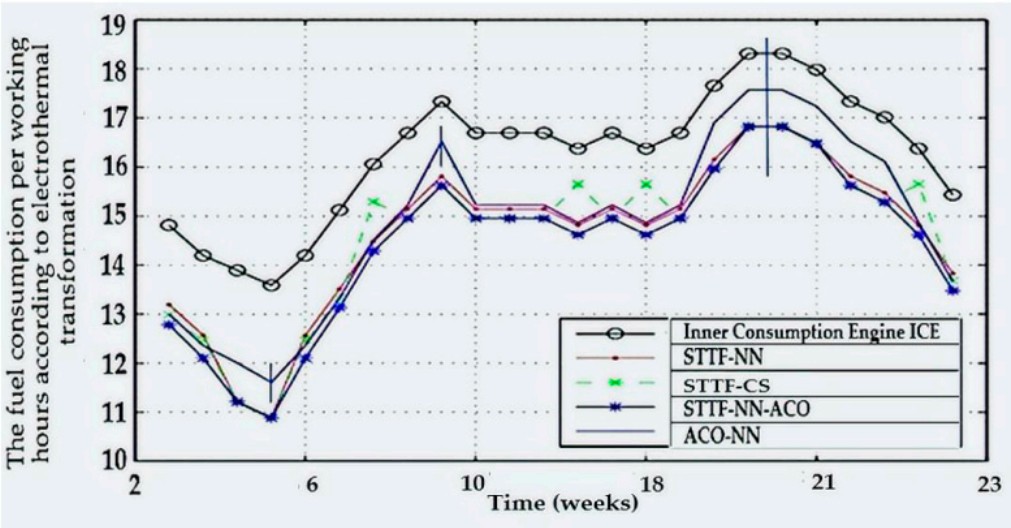

**Figure 22.** The daily fuel consumption for TEG-ICE is affected by the cracks' prediction and reinstalling the thermal conductors in places far away from cracked positions.

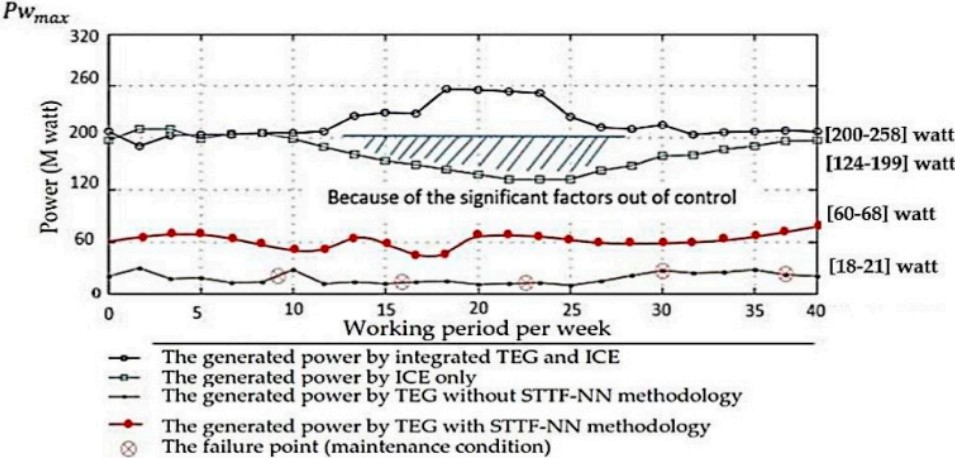

**Figure 23.** The generated electrical power for the diesel and thermoelectric generators through the digital twin simulator through 6241 working hours.

## 5. Conclusions

The main objective of the paper is to track the heat transfer, which is due to the difference in temperature between the hot and cold gaskets via the thermal conductors representing the TEG legs, which must be installed in a safe path on the gasket surface without any obstacles preventing heat transfer (e.g., cracks). The main obstacle is the failure of some positions and their precise intensity, as illustrated in Figures 5–8 on the gasket texture, to install the thermal conductors far away from these positions. When the installation of electro thermal conductors began to measure the amount of heat transfer and generated an output of power during a long working time extended to more than 6241 h. All equations are formulated to serve these two objectives through the digital simulator

twin which tests the efficiency of hybridization of TEG and another ICE generator which is illustrated in Figure 18. The behavior of heat transferred is illustrated in Figure 20a,b, and the prediction efficiency developed in the laboratory as illustrated in Figure 21 by tracking and testing 45 gaskets over 23 weeks (3360 hr.) and training the NN to extract data from the STTF images to enhance the ACO and cuckoo search techniques in predicting the cracks' breeding locations, which gives the model an advantage for working and extracting power for more than 6241 continuous hours. The cracks' diffusion is proportional to the failure of thermal conductors' terminals fixed on the gaskets' slots and formed from the same alloyed material in strip shape and affects the electrically generated power and the transfer efficiency η as illustrated in Figure 22. Identifying the safe track to install the thermal conductor strips on the gasket surface increases the continuous working hours of generating electricity from 5184 to tm: 6241 h (20.39%). Table 6 shows the average deviation from the actual observation values along tm: 6241 working hours via the proposed predicting method and the three others based on the same data extracted from filming the cracks' growth [70–74]. The efficiency of prediction for cracks' positions and their intensity is positively reflected on fuel consumption cost because of the reliance of the vehicle on electricity generated, where Table 5 shows the superiority of STTF-NN-ACO over STTF-CS and Mat-ACO algorithms by 19.43% and 18.95% respectively. However, the interference of extracting the data using the neural network empowers the prediction training to precisely determine the best path for installing the thermal conductivity strips.

**Table 6.** The average deviation from the actual observation values along tm: 6241 working hours.

| Working Hr. | Optimization Prediction Algorithms | Generated Power Per Hour | number of Terminals' Cracks | Fuel Consumed Per Week | Working Weeks | $C_\sigma$: Costs Per Week | Equation (4) Output | Cracks' Position Deviation | Cracks' Intensity mm² | η |
|---|---|---|---|---|---|---|---|---|---|---|
| Native TEG | | 132 KW | 216 | 160 Liter | 23 weeks | $1.026 \times 10^3$ | —— | —— | 9 | 77% |
| Controlling the significant parameters | | 148 KW | 142 | 112 Liter | 31 weeks | $0.826 \times 10^3$ | Along 23 weeks | —— | 6 | 81% |
| *Optimization Improve >* | | +12.15% | −1.33% | −1.83% | +2.52% | −2.09% | —— | ±1.32% | −3.11% | +2.21% |
| 24–3360 | STTF-ACO-NN | 96.72% | 0.42% | 1.03% | 1.06% | 1.04% | 1.0 | 0.16% | 0.16% | 99.84% |
| | Mat-ACO | 88.40% | 0.90% | 2.50% | 2.20% | 2.30% | 1.5 | 0.96% | 2.43% | 98.30% |
| 3361–5184 | STTF-ACO-NN | 97.50% | 0.45% | 1.06% | 1.09% | 1.07% | 1.6 | 0.17% | 0.17% | 99.83% |
| | Mat-ACO | 89.25% | 1.00% | 2.50% | 1.20% | 2.40% | 2.1 | 0.92% | 2.34% | 98.37% |
| 5185–6241 | STTF-ACO-NN | 99.06% | 0.48% | 1.09% | 1.12% | 1.10% | 2.5 | 0.18% | 0.18% | 99.82% |
| | Mat-ACO | 91.29% | 1.10% | 3.50% | 1.20% | 2.40% | 1.6 | 0.92% | 2.34% | 98.37% |
| The Results | | 165.982 | 95.14 | 92.544 | 35.989 | $0.808 \times 10^3$ | | ±1.32% | 4 | 87.81% |

The proposed model inputs fed by STTF with a minimum deviation compared to the power during the actual working conditions are illustrated in Figure 23, which emphasizes that the thermal conductor's efficiency continues for more than 37 weeks, thanks to the support of TEG for ICE generator.

The authors' work in the future is to replace the generator legs with tubes that have engine oil-based nano-fluids absorbed from the engine base, which contains a variety of nanomaterials using a partial model to inspect the thermal aspect of a Brinkman-type nano-fluid composed of molybdenum disulfide (MOS2) and graphene oxide (GO) nanoparticles. These particles are flowing on an oscillating infinite inclined plate and to classify the asymmetrical fluid even when the magnetism, slip boundary conditions, and engine oil sensitive to the Newtonian heating effect were considered. Additionally, future work will test the YbAl3 (Density: ρ = 5.68 Mg·m$^{-3}$) to study the generated power and its ability of service the electrical stations and to charge the EVs.

**Author Contributions:** "Conceptualization, A.M.A. and S.E.; methodology, A.M.A.; software, A.M.A.; validation, A.M.A., L.F.S. and S.E.; formal analysis, A.M.A.; investigation, A.M.A. and L.F.S.; resources, A.M.A. and S.E.; data curation, A.M.A., L.F.S.; writing—original draft preparation, A.M.A. and S.E.; writing—review and editing, A.M.A. and S.E.; visualization, A.M.A. and L.F.S.; supervision,

A.M.A.; project administration, A.M.A. and S.E.; funding acquisition, A.M.A. All authors have read and agreed to the published version of the manuscript.

**Funding:** This project was funded by the Deanship of Scientific Research at Prince Sattam bin Abdulaziz University, award number IF-PSAU-2022/01/22745.

**Data Availability Statement:** http://idealstandarddeviati.wixsite.com/leansixsigma accessed on 10 December 2022.

**Acknowledgments:** The authors extend their appreciation to the Deputyship for Research and Innovation, Ministry of Education in Saudi Arabia for funding this research through the project number IF-PSAU-2022/01/22745.

**Conflicts of Interest:** The authors declare no conflict of interest.

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
