# Peer review of "Building a Digital Twin Simulator Checking the Effectiveness of TEG-ICE Integration in Reducing Fuel Consumption Using Spatiotemporal Thermal Filming Handled by Neural Network Technique"

_processes, doi:10.3390/pr10122701_

Round 1

Reviewer 1 Report

This manuscript treats about building a digital twin model checking the effectiveness of TEG-ICE integration in reducing fuel consumption using spatiotemporal thermal filming handled by neural network technique. I recommend to accept it subjected to the following minor modifications:

- A slight review of the English must be realized. For instance, in line 46 change “doesn’t” to “does not”, in line 51 “weren’t” to “were not”, in line 216 change “couldn’t” to “could not”, etc.

- Improve the quality of figures 6, 7, 21, and 22.

- Review the citation of the references. For instance, in line 81 change “K.M. Winslow et al. (2019) [5]” to “Winslow et al. [5]”, line 170 change “to_Hotta et al. (2018) [21]” to “to Hotta et al. [21]”, page 5 change “Admiral P. et al. 1986 [40], and Richard W. a Hertzbert G. 1996 [41]” to “Admiral et al. [40]; Richard and Hertzbert [41]”, line 428 change “Rodrigo and Gambardella (1997)” to “Rodrigo and Gambardella [xxx]” where xxx is the number in the references section, line 437 change “Mostafa A. et al. (2021) [48]” and do not employ cursive font, to “Mostafa et al. [48]”, line 476 change “Yuanying Gan et al. (2022) [52]” to “Yuanying Gan et al. [52]”, line 481 change “k. R. Kalantari (2020)” to “Kalantari et al. [55]”, line 485 “Yang, X.S., (2010)” to “Yang [56]” and “T. Qiu (2020)” to “Qiu et al. [57]”, line 486, change “F. Han et al (2020)” to “Han et al. (2020)”, line 495 change “M. Khosravy et al. (2020)” to “Khrosravy et al. [61]”, line 531 change “Shen, Z.-G.; Tian, L.-L; Liku, X. (2019) [63]” to “Shen et al. [63]”, line 613 change “(Jiangtao Hong et al., 2018 and Ahmed M. Abed et. Al., 2023) [48, 49]” to simply “[48, 49]”, etc.

- Uniform the units. For instance, in line 177 and 357 the authors use “watts” and in line 251 “W”, in line 673 the authors employ “hrs” and in line 357 “hr”, etc.

- Lines 40 and 55, change “inner” to “internal”.

- Line 67, change “The cause” to “Cause”.

- Line 87, change Co2 to CO2 and write 2 as subscript.

- Line 125, change “Simulator” to “simulator”. Idem in lines 155, 161, 186, etc.

- Lines 212-214, check the font style.

- Line 327, change “Eqns.” to “Eqs.”.

- Line 343, change “Intake” to “intake”, “Exhaust” to “exhaust” and “Merit” to “merit”.

- Line 383, do not numerate that equation as 7a, numerate it as 8 and re-numerate the posterior ones.

- Line 407, change “Digital” to “digital”.

- Line 585, change “Figure 20a” to “Figure (20a)”.

- Line 681, change “Table (7)” to “Table 7”.

- Do not include section 6. The content of section 6 must be merged with section 5. Future work is usually mentioned at the end on the conclusion section.

- Review the format of the references according to the rules of the editorial.

Author Response

Thank you for your notes which have helped in enhancing the manuscript presentation and increase its quality 

Reviewer 2 Report

My opinion about this review paper is positive, it is rich of information and easy to read. This article gives as a complete study on Building a Digital twin model checking the effectiveness of TEG-ICE integration in reducing Fuel Consumption using Spatiotemporal Thermal Filming handled by Neural Network technique. The manuscript is interesting, but the following aspects should be reviewed for the next submission:

(1). Some references are missing in the introduction.

(2). Figures are not clear; their quality must be improved.

(3). The quality of the presentation needs to be improved significantly before considering for publication.

Author Response

(The authors gave the same response as above.)

Reviewer 3 Report

Dear Authors,

Please find attached remarks on your article.

Best regards

Reviewer

Author Response

(The authors gave the same response as above.)
